

# Assessing Future Hydrological Impacts of Climate Change on High-Mountain Central Asia: Insights from a Stochastic Soil Moisture Water Balance Model

Tobias Siegfried[1], Aziz Ul Haq Mujahid[2], Beatrice Sabine Marti[1], Peter Molnar[3], Dirk Nikolaus Karger[4], and Andrey Yakovlev[5]

[1]hydrosolutions GmbH, Zurich, Switzerland
[2]Swiss Federal Institute of Technology, ETH Zurich, Switzerland
[3]Dept. of Civil, Env. and Geomatic Engineering, Swiss Federal Institute of Technology, ETH Zurich, Switzerland
[4]Swiss Federal Institute for Forest, Snow and Landscape Research WSL, Birmensdorf, Switzerland
[5]Freelance, Tashkent, Uzbekistan

*Correspondence to*: Tobias Siegfried (siegfried@hydrosolutions.ch)

**Abstract.** We use a new set of data available to compute 21st century climate impacts on the hydrology of 221 catchments in high-mountain Central Asia. For each of these subcatchments, a parsimonious steady state stochastic soil moisture water balance model was set up and the partitioning of available water from precipitation into runoff and evaporation computed for different climate futures using the Budyko framework. Climate change sensitivity coefficients are analytically derived for the first time using the total differential method. Relative changes in discharge for three future periods 2011 – 2040, 2041 – 2070, and 2071 – 2100 were computed in relation to the baseline period from 1979 – 2011. For the baseline observation period, climate data from a global high-resolution climatology data set (CHELSA V21) were used to extract mean daily subcatchment-specific temperature and precipitation values. Data from the coupled model intercomparison project phase 6 (CMIP6) were used to compute catchment mean future climate data using 4 GCM models with 4 scenario runs each. CMIP6 data were bias corrected with CHELSA V21 observation data. For the spatial distribution of soil parameters, different global products were utilized. The robustness of the soil water balance model results was assessed using a comprehensive sensitivity analysis in relation to variations of these soil parameters over typically observed ranges for each subcatchment.

The analysis of climate change suggests increasing precipitation over the three periods (+4.44%, +5.89%, and +8.51% relative increases in median total precipitation averaged over subcatchment and scenarios). Median values of temperatures changes between periods relative to the baseline are +1.33 °C, +2.44 °C, and +3.55 °C. Results of the hydrological soil water balance model runs suggest a median increase of discharge of +4.71%, +7.44% and +10.87% for the corresponding periods. This is a strong indication of a wetter and hotter future in Central Asia, relative to today's hydroclimate. Modelling results suggest that decreasing contributions from glacier melt over the course of the 21st century will be offset by increases in discharge consistently throughout the region, despite increasing potential evapotranspiration. Increases in relative discharge will be most pronounced in the Afghan Murghab-Harirud basin and in the Amu Darya. Changes in precipitation characteristics in terms of



frequency and event depth also indicate possible impacts on hydrological extremes which remains a heavily under researched topic in Central Asia.

**Short Summary.**
The study assesses the potential impacts of climate change on hydrology in 221 subcatchments in high-mountain Central Asia. The study uses a stochastic soil moisture water balance model to calculate how precipitation is partitioned into runoff and evaporation for different climate scenarios. The results suggest that there will be increasing precipitation and temperatures in the region, leading to a median increase in discharge of up to 10.87% over the 21st century, depending on the particular climate scenario under consideration. The study also identifies possible impacts on hydrological extremes and highlights the need for further research in this area.

**Acknowledgement.** This project has received funding from the European Union's Horizon 2020 research and innovation programme under grant agreement No 101022905 (Hydro4U Project, see [www.hydro4u.eu](http://www.hydro4u.eu)). We thank the anonymous reviewers for helpful suggestions and comments. Dr. Silvan Ragettli is acknowledged for helpful discussions.

**Author Contribution.** TS designed the research, derived the analytical expressions of the climate sensitivity indices, and implemented the model. AM implemented the evaporation model and carried out different sensitivity model runs as part of an MSc Thesis. BM contributed the glacier analysis. PM provided critical feedback for the entire research. DK contributed CHELSA high-resolution climate data. AY supported TS in finding gauge locations and mapping subcatchment boundaries. AY also engaged in critical discussions about the individual findings in the large river basins and provide valuable feedback on the methodology and results.



## 1 Introduction

Climatic change is expected to significantly impact the regional hydrological systems in semi-arid Central Asia (Sorg et al., 2012; Unger-Shayesteh et al., 2013; Su et al., 2022; Li et al., 2020a; Siegfried et al., 2012; Gulakhmadov et al., 2020; Zhang et al., 2020; Khanal et al., 2021). The resultant changes in water availability across seasons and at interannual scales will have large impacts on economic sectors (Shaw et al., 2022), on human livelihoods (Xenarios et al., 2019; Reyer et al., 2017) and on ecosystems (Chen et al., 2022), and possibly constitute a security risk (Bernauer and Siegfried, 2012). Robust projections on

climate impacts are important for adaptation and mitigation alike, for example by informing on effective renewable energy sector development strategies over the course of the 21$^{st}$ century as estimates about changes in the hydropower production potential (Laldjebaev et al., 2021) can support investment decisions that are key towards a green energy transition in the region.

In this context, a good understanding of the impacts of atmospheric warming on land ice and snow cover dynamics in the

Central Asian high mountain areas has been obtained with recent advances in research and novel sources of data (Rounce et al., 2020; Hugonnet et al., 2021; Immerzeel et al., 2020; Huss and Hock, 2018). Changes in the discharge seasonality, including a decreasing ratio between warm and cold season runoff volumes and the shift of warm season peak discharge towards boreal spring, and the timing of peak water from glacier ablation are expected to be directly related to the magnitude of warming and the pace at which it occurs (Rounce et al., 2023). Many of these changes are already observed nowadays (Li et al., 2020b).


Hand in hand with these developments, climate change will also cause significant ecohydrological impacts. A rising climatic snowline and widespread permafrost thawing creates new ecological niches in the mountainous places that can be populated under a milder climate and under adequate supplies of water (Calleja-Cabrera et al., 2020; Hauser et al., 2022). The associated changes in vegetation species distribution will be difficult to observe and to quantify. Yet, they are of utmost importance due

to the critical role that soils play in the partitioning of available water into evaporation and runoff as well as deep infiltration (Lian et al., 2021; Porporato et al., 2004; Porporato and Yin, 2022).

Such impact studies in the semi-arid Central Asia region are limited by the paucity of in-situ data available for hydrological model calibration, and validation. While new data on historic discharge do change this situation to a certain extent (see, e.g.,

Beatrice Marti, 2023), the geographic complexity of the basins and the rivers and their tributaries draining the Hindukush, the Pamirs, Gissar-Alay and Tien Shan mountain ranges, with contributions from Afghanistan, Kazakhstan, Kyrgyzstan, Tajikistan, Turkmenistan and Uzbekistan, is immense. Even at the time of the highest observational density in the 1960s, the local hydrometeorological observation network was never adequate to account for highly specific local hydroclimatological conditions, changing from one catchment to the next, and also across the altitudinal zones (Shults, 1965). It is thus not further

surprising that such comprehensive regional hydrological impact study is missing so far.



Addressing this shortcoming and by using a simple stochastic soil water balance model in combination with the dimensionless Budyko framework, we derive the climate sensitivity of discharge in relation to relative changes in precipitation depth and frequency, evaporation, and soil water available for evaporation under the assumption of changes from one steady state *ex ante* to another *ex-post*. We then apply this model to 221 subcatchments in the zone of runoff formation in high mountain Central Asia for which the relevant data on discharge, soil, and root zone characteristics, and historic as well as future projected climatologies were available. Using these data, subcatchment specific climate responses were computed in terms of relative changes in specific mean annual discharge for different future periods and under different representative concentration pathways end emission scenarios. To arrive at robust climate impact estimates, subcatchment level results are aggregated over the basins and aggregate responses are reported and discussed.

Section 2 provides an overview of the geography of the Central Asia region. The stochastic soil water balance model is presented and the climate sensitivities analytically derived (see also Appendix A). Discharge, climate, and soil data as well as data on glacier melt are presented and discussed. Section 3 discusses 21st century climate change in the region as derived from a diverse set of CMIP6 model runs and quantifies climate impacts on the hydrological system. Sensitivity analysis is carried out in relation to different rooting zone depth products. Section 4 concludes with a discussion on the potential implications of the projected changes on sectors, livelihoods, and ecology.

## 2 Methods and Materials

### 2.1 Geographic Overview

Figure 1 shows a map of the study region. The colored area covers a total of 423'099 km$^2$ spreading over 221 subcatchments with a median area of 1'427.6 km$^2$ (91 catchments are smaller than 1'000 km$^2$) for which discharge data are available. It includes parts of Kazakhstan (KAZ), Kyrgyzstan (KGZ), Tajikistan (TJK), Turkmenistan (TKM) and Uzbekistan (UZB). The colors demark the large basins to which the corresponding subcatchment belongs to. They drain the Hindukush, Pamir, Gissar-Alay, and Tien Shan Mountain ranges.

Individual basin areas are as follows: Amu Darya 230'782 km$^2$, Chu-Talas 15'266 km$^2$, Issik-Kul 5'924 km$^2$, Murghab-Harirud 63'415 km$^2$, and Syr Darya 107'744 km$^2$. The mean subcatchment elevation is 2'791 meters above mean sea level (masl), with a minimum mean subcatchment elevation of 877 masl and a maximum mean subcatchment elevation of 4'641 masl. The catchments highlighted in Fig. 1 cover most of the high-mountain zone of runoff formation in semi-arid Central Asia.

Mean annual specific discharge values are 382 mm in Amu Darya (norm precipitation P: 819 mm, mean temperature T: -0.3 °C, aridity index $\phi$: 1.16, evaporative fraction $\varepsilon$: 53.4%), 329 mm in Chu-Talas (P: 1012 mm, T: -2.1 °C, $\phi$: 0.65, $\varepsilon$: 67.5%), 459 mm in Issyk-Kul (P: 885 mm, T: -4.7 °C, $\phi$: 0.68, $\varepsilon$: 48.1%), 67 mm in Murghab-Harirud (P: 589 mm, T:





6.8 °C, $\phi$: 2.05, $\varepsilon$: 88.7%), and 415 mm in Syr Darya (P: 974 mm, T: -0.1 °C, $\phi$: 0.82, $\varepsilon$: 57.4%). All values are long-term
averages in time and averaged in space over all corresponding subcatchments (see Section 2.3 for more information on data and sources). Thus, approximately 42% of the total water supplies of the zone of runoff formation, on average, discharge to the arid plains where most of this water is subsequently used consumptively in irrigation and returned to the atmosphere via evapotranspiration.

**2.2 Hydrological Modeling**

The choice of a hydrological modeling approach is informed by data availability. Different approaches for studying climate impacts in Central Asia have been followed over the past decade or so (Siegfried et al., 2012; Huang et al., 2022; Didovets et al., 2021; Shannon et al., 2022; Hu et al., 2021; Loodin, 2020). These modeling studies feature various levels of complexity, depending on the modelling scale and the locations. Often, they focus on a few selected representative catchments from where data are available over a longer period for model setup, calibration, and validation of dynamical models.


The focus of the regional study presented here is on the entire zone of runoff formation in southern and south-eastern high mountain Central Asia (Fig.1). This large spatial focus significantly restricts the set of modeling approaches. Thus, using Budyko type parsimonious modeling, we present a simple distributed regional approach to quantify climate impacts on norm specific discharge in 221 tributary catchments of the large basins Amu Darya, Chu-Talas, Issyk-Kul, Murghab-Harirud, and
Syr Darya.

The general water balance of a subcatchment shown in Fig. 1 can be written as

$$\Delta S = P - E - Q \tag{1}$$


where $\Delta S$ is the net soil water storage change in millimeter (mm), $P$ is precipitation in mm, $E$ is evaporation in mm, and $Q$ is specific discharge in mm. This definition of evaporation encompasses evaporation from inside leaves (transpiration), evaporation from bare soils, evaporation from intercepted precipitation, evaporation from open water surfaces, and finally, evaporation over ice- and snow-covered surfaces (Miralles et al., 2020).


The Porporato simplified stochastic soil moisture dynamics model (Porporato et al., 2004) describes the catchment scale water balance and its components in Eq. 1 in a parsimonious way at daily time scales as

$$w_0 \frac{dx(t)}{dt} = P_d(t) - E_d\big(E_m, x(t)\big) - Q_d(x(t), t) \tag{2}$$




where $x(t)$ is the state variable, i.e., the effective relative soil moisture where $x(t) = (s(t) - s_w)/(s_1 - s_w)$ with $s(t)$ (-) being the relative soil moisture and $s_1$ (-) is the threshold above which rainfall in excess of available storage capacity is directly converted into runoff $Q_d$ (mm) under well-watered conditions or lost via deep percolation and drainage, and $s_w$ (-) is the wilting point of the soil. $w_0 = (s_1 - s_w) \, n \, Z_r$ (mm) is the maximum soil water storage available for evaporation with $Z_r$ (mm) being the rooting zone depth and $n$ (-) being the porosity of the soil column. We refer to this model as PSM model (Porporato et al., 2004; Daly et al., 2019; Porporato and Yin, 2022).

$P_d$ (mm/day) is the daily rainfall modeled as stochastic rare-event Poison process with constant frequency $\lambda$ (1/day) delivering a daily rainfall event depth that is drawn from an exponential distribution with a mean $\alpha$ (mm). The long-term average annual rainfall over a catchment can be expressed as $P = \alpha \lambda n_d$ with $n_d$ being the number of days per year (Daly et al., 2019).

Finally, in a departure from the original formulation by Milly (1994), the PSM model assumes that actual daily evaporation $E_d$ (mm/d) depends linearly on the potential evaporation $E_m$ (mm/d), i.e., $E_d = E_m x(t)$, thus accounting for a reduction of evaporation under increasing water stress (Daly et al., 2019). Under the simplifying assumption that $E_m$ is not dependent on time, i.e., shows no seasonality, an average annual rate of *potential* evaporation $E_p$ can be computed as $E_p = E_m n_d$ (mm/a). In the following, the *actual* annual evaporation is denoted as $E$. The annual average discharge $Q$ (mm) can be computed in a similar fashion from daily values, i.e., $Q = Q_d n_d$.

Under the assumption of $w_0 dx(t)/dt = 0$ in Eq. (2), the long-term partitioning of rainfall into evaporation $E$ and runoff $Q$ averaged over a catchment domain at the annual timescale can be described as

$$0 = P - E - Q \tag{3}$$

With constant $\alpha$ and $\lambda$ and a linear dependence of evaporation on soil moisture, Daly et al. (2019) showed that the normalized steady-state water balance equation can be analytically solved as

$$\frac{E}{P} = 1 - \frac{\Phi \, \gamma^{\gamma/\Phi - 1} e^{-\gamma}}{\Gamma\left(\frac{\gamma}{\Phi}\right) - \Gamma\left(\frac{\gamma}{\Phi}, \gamma\right)} \tag{4}$$

with $\phi = E_p/P$ (-) being the aridity index and $\gamma = w_0/\alpha$ (-) being the storage index of a basin. $\gamma$ describes soil water under well-watered conditions normalized by the rainfall depth and determines how much water is retained by a catchment relative to average precipitation event depth. In the types of models studied here, an increase in $\gamma$ is associated with increased losses





of soil moisture to evaporation (e.g., Fig. 1 in Porporato et al. (2004)). $\Gamma(.)$ is the gamma function and $\Gamma(.,.)$ is the incomplete gamma function (Abramowitz Milton, 1964). Using Eq. (3), the long-term norm discharge as per Eq. (4) can be written as

$\quad Q = P\,g(\Phi,\gamma)$ (5)

with

$$g(\Phi,\gamma) = \frac{\Phi\,\gamma^{\gamma/\Phi-1}e^{-\gamma}}{\Gamma\left(\frac{\gamma}{\Phi}\right)-\Gamma\left(\frac{\gamma}{\Phi},\gamma\right)}$$ (6)


Contrary to the model formulated by Budyko (1974), where the norm discharge is described by

$$Q = P\left(1 - f(\Phi)\right)$$ (7)

and

$$f(\Phi) = \sqrt{\left(\Phi\,\tanh(\Phi^{-1})\,(1 - e^{-\Phi})\right)}$$ (8)

the steady-state solution of the PSM model as shown in Eq. (5) thus considers not only the influence of the aridity index on
the partitioning of precipitation into runoff and evaporation but also considers differences in physical catchment characteristics via the dependency on the storage index $\gamma$.

The model shown in Eq. (4) is in this sense a parametric Budyko curve with $\gamma$ being the additional parameter. However, unlike in the many parametric extensions of the original Budyko framework (see e.g. Reaver et al. (2022)) for a comprehensive
overview of the vast literature in this regard), the additional parameter is directly derived from physical catchment characteristics (mean depth of rainfall events and rooting depth) and therefore does not simply arise out of mathematical necessity to be able to explain a vertical scattering of the data observations around the semi-empirical original Budyko curve.

## 2.3 Quantification of Climate Change Impact

Using Eq. (5), first order changes in the discharge $dQ$ as a function of changes in climate ($dP$, $d\phi$) and soil properties $d\gamma$ can be expressed as



$$dQ = g(\Phi, \gamma)dP + P\frac{\partial g(\Phi, \gamma)}{\partial \Phi}d\Phi + P\frac{\partial g(\Phi, \gamma)}{\partial \gamma}d\gamma \qquad (9)$$

When expanding $dP$ and $P$, $d\phi$ and $\phi$, $d\gamma$ and $\gamma$ in terms of $\alpha$, $\lambda$, and $E_p$ in Eq. (9) and by collecting the terms, we get for relative changes

$$\frac{dQ}{Q} = \sigma_{E_p}\frac{dE_p}{E_p} + \sigma_{w_0}\frac{dw_0}{w_0} + \sigma_\alpha\frac{d\alpha}{\alpha} + \sigma_\lambda\frac{d\lambda}{\lambda} \qquad (10)$$

where the relative changes are computed as $dQ/Q = (Q(p,s) - Q(0))/Q(0)$, with $p$ in $Q(p,s)$ denoting the target future period and $s$ refers to the Shared Socio-Economic Pathway and Representative Concentration Pathway scenario (SSP, see also Section 2.4 for more information). $Q(0)$ is the baseline norm discharge. The baseline identifier is dropped in the terms denoting the relative changes for clarity.

The $\sigma_E$, $\sigma_{w_0}$, $\sigma_\alpha$, and $\sigma_\lambda$ can be regarded as sensitivity coefficients. Roderick and Farquhar (2011) derived in a similar manner sensitivity coefficients for the Choudhury Budyko equation. These sensitivity coefficients are functions of the baseline norm discharge $Q(0)$, and the baseline climate conditions $\alpha$, $\lambda$, and $E$ as well as the maximum soil water storage available for evaporation $w_0$ under the baseline conditions.

After a somewhat tedious derivation (see Appendix A), it can be shown that the following holds for the individual sensitivity coefficients

$$\sigma_{E_p} = \frac{E_p}{Q}\frac{\partial g(\Phi, \gamma)}{\partial \Phi} = 1 - \chi \qquad (11)$$

$$\sigma_{w_0} = \frac{\lambda w_0}{Q}\frac{\partial g(\Phi, \gamma)}{\partial \gamma} = \chi + \frac{\gamma(1 - g(\Phi, \gamma))}{\Phi} - \gamma - 1 \qquad (12)$$

$$\sigma_\alpha = 1 - \sigma_{E_p} - \sigma_{w_0} = 1 + \gamma - \frac{\gamma(1 - g(\Phi, \gamma))}{\Phi} \qquad (13)$$

$$\sigma_\lambda = 1 - \sigma_{E_p} = \chi \qquad (14)$$

with $\chi = e^\gamma \, {}_2F_2\left(\frac{\gamma}{\phi}, \frac{\gamma}{\phi}; \frac{\gamma}{\phi} + 1, \frac{\gamma}{\phi} + 1; -\gamma\right) g(\phi, \gamma)$ with ${}_2F_2(.)$ being the hypergeometric function (Abramowitz Milton, 1964).

With Eq. (11) – (14) in combination with Eq. (10), we can analytically compute subcatchments-specific relative changes in discharge as a function of changes in climate factors and soil properties for future periods of interest. More precisely, these are



$$\frac{dQ}{Q} = (1 - \chi)\frac{dE_p}{E_p} + \left(\chi - \sigma_{w_0}\right)\frac{d\alpha}{\alpha} + \chi\frac{d\lambda}{\lambda} + \sigma_{w_0}\frac{dw_0}{w_0} =$$

$$(1-\chi)\frac{dE_p}{E_p} + \left(-\frac{\gamma}{\phi}\big(1 - g(\phi, \gamma)\big) + \gamma + 1\right)\frac{d\alpha}{\alpha} + \chi\frac{d\lambda}{\lambda} + \left(\chi + \frac{\gamma}{\phi}\big(1 - g(\phi, \gamma)\big) - \gamma - 1\right)\frac{dw_0}{w_0} \tag{15}$$


See Appendix S.1 for more information on the derivation of these results.

The PSM model does not represent cryosphere processes. For example, current and future glacier melt contributions cannot be accounted for. This is critical in high mountain areas close to areas with large glaciation where important percentages of

discharge are from glacier balance and imbalance ablation (Miles et al., 2021). We are cognizant of this model deficit and thus account for glacier contributions in terms of per period specific changes via available existing data on recent observed and future modelled glacier melt data (Rounce et al., 2023). We further assume that snow melt and accumulation dynamics are irrelevant at the annual time scale.

Thus, for each subcatchment after having derived the delta change values, we are first computing changes in absolute terms by applying the delta changes to the net baseline discharge values $Q_{net}(0) = Q(0) - Q_G(0)$, i.e., to subcatchment specific discharge values where specific glacier discharge has been removed. Where $Q(0) < Q_G(0)$, we set $Q_G(0) = Q(0)$ while assuming that all the observed discharge is from glacier melt in the subcatchment. The future discharge, including the changes from the PSM model and the changes from glacier dynamics predicted by Rounce et al., 2023, can then be computed for all

subcatchments individually to obtain the future discharge:

$$Q(p) = (1 + \Delta)\, Q_{net}(0) \; + Q_G(p) \tag{16}$$

where $\Delta$ is a placeholder for the corresponding target period and scenario specific delta change coefficient as computed by Eq.

(15) above. The relative changes, including from glacier melt, are then computed via $\big(Q(p) - Q(0)\big)/Q(0)$.

## 2.3 Discharge and Climate Data

Long-term annual norm discharge data from 299 mountainous gauging stations from Kyrgyzstan, Kazakhstan, Uzbekistan and Tajikistan could be obtained from the local Hydrometeorological Organizations, public reports and the Soviet compendia

Surface Water Resources, Vol 14 Issues 1 and 3 (Main Directorate of the Hydrometeorological Service under the Council of Ministers of the USSR (original: Главное управление гидрометеорологической службы при совете министров СССР), 1969, 1971). Norm discharge data and the location of gauges located in the Afghan part of the Amu Darya River basin were obtained from Olson and Williams-Sether (2010). Variable observation periods at the individual gauging stations for the



computation of the norm discharge were used with data spanning the years 1910 – 2011. Where discharge time series data are
available, trend direction of the mean annual discharge measured with the Sen's slope are shown with corresponding color
coding (Fig.1).

As there are no consistent geolocated records of gauging stations available, all gauging stations were manually located in a
Geographic Information System (GIS). In the former Soviet region, parts of gauging station names often consisted of the
village names where they are (were) located. For this reason, we developed a workflow to manually match station location
names with village names found on the relevant Soviet Topographic maps (1:200'000) from the corresponding region and
along the corresponding rivers. The topographic maps were downloaded from https://maps.vlasenko.net and subsequently
manually georeferenced in QGIS (QGIS.ORG, 2020). Gauge locations where then inferred by visually inspecting high
resolution optical remote sensing imagery by locating obvious measurement locations, such as bridges or installations that
would allow for cross-section measurements of water depth and velocity.

For each gauge, the contributing watershed area was delineated in R with the WhiteboxTools v2.0.0 (Lindsay, 2016). The
NASA SRTM digital elevation model 1 Arc-second (30 m) global product was used as a digital elevation model (DEM)
(NASA JPL, 2013a). We limit our study to subcatchments larger than 200 km$^2$ which is an arbitrary chosen cutoff value to
avoid spurious statistics when sampling finite resolution raster values from comparatively small polygons. This brings down
the number of catchments to 221.

CHELSA V21 global daily high-resolution precipitation and temperature climatologies were processed over the Central Asian
domain and norm 2 meters (m) surface temperatures and total norm annual precipitation (mm) values computed for the
catchments for the period 01-01-1979 until 31-12-2011 (Karger et al., 2021, 2017). The CHELSA V21 product is corrected
for snow undercatch in the high elevation ranges and thus can better represent actual high mountain precipitation than other
available global climatologies (Beck et al., 2020). For the computation of subcatchment-specific mean raster statistics, we use
the exactextractr R Package (Daniel Baston, 2022) throughout.

We define the period from 1979 to 2011 as the baseline climate period p0 against which we measure change. Figure 2 shows
mean aridity index values (left plate) and norm precipitation values (right plate) for this baseline period and all subcatchments.
The low-lying highly arid catchments located in the southern and south-western part of the study area together with the high
elevation intramountain Naryn and Pamir block subcatchments are characterized by high aridity and low norm precipitation
values.

Figure 3 shows precipitation characteristics for each of the subcatchments and the baseline period. The left plate shows mean
event depth values $\alpha$ (mm) whereas the right plate shows subcatchment-specific event frequencies $\lambda$ (a$^{-1}$) for the baseline



period. Note, $P = \alpha\,\lambda$ where $P$ is the mean annual precipitation. For the computation of $\alpha$ and $\lambda$, a precipitation threshold of 1 mm per day was defined above which a wet day is counted.


The plates in Fig. 3 confirm our understanding of the hydroclimatological system in high-mountain Central Asia with the western flanks of the large mountain ranges receiving the bulk of the precipitation during the winter months via moisture transport with the westerlies. The data suggest that surrounding Lake Issik-Kul there are anomalously high precipitation frequencies. These might be directly related to the local moisture recycling from the vast lake's surface.


To study the 21$^{st}$ century climate impacts in the region, three future climate periods are defined as follows

- **period p1**: 2011 - 2040,
- **period p2**: 2041 - 2070, and
- **period p3**: 2071 - 2100.

Future global circulation model (GCM) daily precipitation and temperature time series from the CMIP6 Coupled Model Intercomparison Project Phase 6 were downloaded from the Copernicus Climate Change Service (C3S) Climate Data Store. Data from the following four models were downloaded and processed: UKESM1.0-LL (Tang et al., 2019), IPSL CM6A-LR (Boucher et al., 2018), MRI-ESM2-0 (Yukimoto et al., 2020), and GFDL-ESM4 (Krasting et al., 2018). GCM selection followed the one from the Intersectoral Impact Model Intercomparison Project (ISIMIP). For more details, see

https://www.isimip.org/documents/413/ISIMIP3b_bias_adjustment_fact_sheet_Gnsz7CO.pdf.

The following four Shared Socio-Economic Pathway (SSP) and Representative Concentrations Pathway (RCP) scenarios are investigated: SSP1 RCP 2.6, SSP2 RCP 4.5, SSP3 RCP 7.0, and SSP5 RCP8.5 (Vuuren et al., 2011; Riahi et al., 2017). For brevity, we refer to these as SSP1-26, SSP2-45, SSP3-70, and SSP5-85 scenarios throughout the text.


The GCM daily climate model raster fields of surface temperature and precipitation were used to compute mean daily time series over the subcatchments. Using the historical GCM simulations from 1979 – 2011 in comparison with the CHELSA V21 daily data, the individual timeseries were bias corrected in R for each scenario and each GCM separately using quantile mapping available via the qmap R Package (R Core Team, 2022; Gudmundsson et al., 2012). Figure 4 shows the scenario and

pathway dependent changes of the distributions of the mean temperature and precipitation values over the 221 subcatchments, ensembled over the 4 GCM.

For the computation of the potential evaporation over the baseline period and the future periods p1-p3, daily GCM temperature time series were used to compute period averaged $E_p$ values using Jensen-Haise and McGuiness model formula as detailed in

Oudin et al. (2005), with





$$E_p = \frac{R_e}{l\varrho} \frac{T_a + k_2}{k_1}, \text{ if } T_a + k_2 > 0, \text{ else } E_p = 0 \qquad (17)$$

where $R_e$ is extraterrestrial radiation (MJ m$^{-2}$ day$^{-1}$), $l$ is the latent heat flux (MJ kg$^{-1}$), $\varrho$ is the density of water (kg m$^{-3}$), and

$T_a$ is the mean daily air temperature in °Celsius. (Oudin et al., 2005) showed that by selecting $k_1 = 100$ and $k_2 = 5$, Eq. (15) provides input to rainfall-runoff models that were well-performing.

For each subcatchment, we compare subcatchment-averaged CHELSA V21 Penman $E_p$ values (Karger et al., 2017, 2020) with values obtained from the application of Eq. (17) with $k_1 = 100$ and $k_2 = 5$ and ensembled over the GCMs. The left plate

in Fig. 5 shows the results. With this method and when compared to the Penman values for the baseline period, $E_p$ appears to be underestimated by at least a factor of 2 with Eq. (17). Using quantile mapping bias correction, we correct the Oudin $E_p$ values with the "observations" from the CHELSA V21 data set for the baseline period (right plate in Fig. 5) and all future periods p1 – p3.

**2.5 Soil Data**

For the spatial distribution of soil parameters over the subcatchments, different global dataset are used. For the soil porosity $n$, the WCsat layer is taken from the HiHydroSoil V2.0 data (Simons et al., 2020). Figure B1 in the Appendix shows the map.

Subcatchment averaged rooting zone depth values were retrieved from three global data set (Fan et al., 2017; Stocker et al.,

2021; Yang et al., 2016). The corresponding maps are shown in the Appendix in Fig. B2 – Fig. B4. Existing outliers in the raster data sets were corrected by restricting $Z_r$ depth values to not exceed 50 meters. Missing data at the subcatchment level in the (Yang et al., 2016) data set (see Fig. B4) were filled using averaged values from neighboring subcatchments. The resulting histograms in Fig. 6 show the product specific distribution of effective $Z_r$ values for the 221 catchments.

As is visible, large differences in the estimates of $Z_r$ across these products exist. Over the study domain, the median rooting depth of the (Fan et al., 2017) data equals 1'289 mm whereas for data from (Stocker et al., 2021) it is 3'127 mm and for the (Yang et al., 2016) it is 185 mm (Fig. 6). Resulting implications will be discussed further in Section 3.3. For the moment, we assume that the data by Fan et al. (2017) represent a good intermediate compromise product for the study region.

The data by Fan et al. (2017) are model-derived *maximum* rooting depth values. We assume that this maximum rooting depth corresponds to D99 levels, i.e., to the depth above which 99% of the root biomass is located. We also assume that both the data by Stocker et al. (2021) and by Yang et al. (2016) correspond to D99 values of $Z_r$.



Averaging D99 values over subcatchments to arrive at an 'effective' rooting depth $Z_r$ for the model presented in Eq. (2) might

lead to an overestimation of the maximum soil water storage that is available for evaporation $w_0$. We rescale the D99 values

to D66 values to account for the fact that vegetation is extracting water much closer to the surface than at D99 levels and are

using conversion factors reported in Table A1 in Hauser et al. (2022). The table has been reproduced in a modified way in the

Appendix S.3 in Table A1. For the computation of $w_0$ and $\gamma$, we then use these D66 $Z_r$ values (median value of 367 mm over

the 221 subcatchments). We discuss the importance of the choice of $Z_r$, $w_0$, and $\gamma$ in the PSM model and the sensitivity of our

model results in relation to these assumptions in the Section 3.3 below. Figure 7 shows the geographic distribution of the D66

$w_0$ and D66 $\gamma$ values.

Finally, data from Simons et al. (2020) were taken for $s_1$ (WCsat data) and $s_w$ (WCpF4.2 data) (Fig. B1 and Fig. B5 in

Appendix). We use these together with porosity values to compute $w_0$ from $Z_r$. The results are shown in the left plate of Fig.

7. It should be noted that Stocker et al. (2021) also provides global data on rooting zone water storage capacity $S_0$, apart from

$Z_r$ values. This $S_0$ is corresponding to $w_0$ from the PSM model (see Section 3.3 where the sensitivity of model results to these

data are studied).

Unlike in the case of the climate variables, model-based projections of climate-induced changes to the maximum soil water

storage available for evaporation $w_0$ are not available. This is critical since it is well-known that the dominant drivers of root

zone distribution values $Z_r$ (on which $w_0$ linearly depends) are influenced by plant functional types, water availability and

temperature and changes therein (Tumber-Dávila et al., 2022; Schenk, 2008; Nippert and Knapp, 2007; Fan et al., 2017;

Jackson et al., 1996; Ellis et al., 2010; Edgeworth et al., 2015). Significant gaps in the understanding of deep root systems as

well as the changes to root zone distributions and the timescales involved persist and insufficient parameterizations of these

distributions impair the accuracy of the projections of changes (Schenk, 2008; Zeng, 2001).

As Donohue et al. (2012) emphasize, despite the fact that climate-related links to $Z_r$ are highly uncertain, incorporating root

zone depth and its dynamics into the PSM model framework provides a way to put these dynamics into the context of other

climate-induced changes contributing to impacts on discharge. We parametrize a very simple climate root zone depth model

to account for these dynamics.

In the recent study by Hauser et al. (2022), global rooting depth changes as a function of human activity related to land

conversion due to agricultural expansion on the one hand and natural changes due to a changing climate on the other are

discussed. The former process replaces natural deep root systems with shallow roots, as has for example likely happened in

the Central Asian irrigated oases since the dawn of the irrigation age (see e.g. Carmi et al. 1992).



Increasing temperatures and atmospheric $CO_2$ concentrations are linked to root extensions (Iversen, 2010; Hu et al., 2018). With strongly increasing temperatures in the Central Asia mountainous region (left plate Fig. 4) and the increasing atmospheric moisture supply there (right plate Fig. 4), it can be envisioned that ecosystem ranges increase across altitudinal zones due to these combined temperature and precipitation effects. In other words, more deeply rooted vegetation becomes more abundant in places where growth would have been unlikely during the baseline period.

In this study, we focus exclusively on the runoff generating processes in the regions outside past agricultural expansions. Due to the hilly and often steep terrain in mountainous subcatchments, it is unlikely that significant large scale land conversion via agricultural expansion will be observed there in the future under climate scenarios. Thus, we do not assume a loss of rooting depth due to such anthropogenic activities there in the future and focus solely on the effect of an increasing rooting depth due to a warmer (and wetter) climatic future in the high-mountain region.

Figure 8 shows the dependency of D66 $Z_r$ reported by Fan et al. (2017) on norm baseline temperature values and on mean subcatchment elevations. Using a simple linear model, we can establish a functional relationship between temperature and rooting depth. As the GCM models give us a climate-driven temperature response, we can translate this response into a rooting depth dynamic under the assumption that a warm, low-lying catchment with a well-established rooting zone today will have a higher lying (in terms of elevation) counterpart with similar root depth characteristics in the future. Hence, for the computation of relative climate-induced changes $dw_0$, we presume

$$dw_0 = (s_1 - s_w)\, n\, dZ_r = (s_1 - s_w)\, n\, \beta\, dT \tag{18}$$

with $\beta$ being the slope parameter of the linear model $Z_r = \alpha + \beta\, T$ where $\beta = 155.7$ mm/°C describes the best fit slope for the data (see also right plate of Fig. 7).

For the computation of $\chi$ and $g(\phi, \gamma)$, as detailed in Section 2.3, numerical issues arise when computing these terms using a machine precision environment. For example, the R package hypergeo (Hankin, 2016) shows significant numerical instabilities in the evaluation of $\chi$. We address this issue by computing the numerical values for each combination of aridity index and basin storage index using Mathematica's adaptive precision feature. $\chi$ and $g(\phi, \gamma)$ are computed with a precision of 200 digits (Version 13.2, Wolfram Research Inc., n.d.). For the computation of the hypergeometric function ${}_2F_2(.)$, we are using the Mathematica function `HypergeometricPFQ[]`. Equipped with $\chi$ and $g(\phi, \gamma)$, we can then easily compute all climate sensitivity indices as per Eq. (11) – (14).





### 2.6. Data on Glacier Melt

Cryosphere processes related to glaciers, permafrost, and snow are not represented in the PSM model. To account for the impact of climate change on the glacier contribution to basin runoff, per-glacier monthly projections of 17'993 glaciers located within the territory of the 221 basins were retrieved from Rounce et al. (2022). Glacier runoff $Q_G$ from the year 2000 to the end of the century were aggregated by subcatchment and averaged over the corresponding periods (Rounce et al., 2023). The monthly runoff data assuming a fixed gauge at a glacier terminus that does not move over time were used (parameter

glac_runoff_fixed_monthly in [https://nsidc.org/sites/default/files/documents/user-guide/hma2_ggp-v001-userguide_0.pdf](https://nsidc.org/sites/default/files/documents/user-guide/hma2_ggp-v001-userguide_0.pdf), Page 4, last accessed January 31, 2023).

The glacier runoff projections have been calculated using the glacier-specific PyGem-OGGM models forced with CMIP6 climate models with the four emission scenarios SSP1-26, SSP2-45, SSP3-70, and SSP5-85. For each subcatchment, we extract

glacier discharge values averaged over all 12 GCM runs available. Table 1 shows a comparison of mean specific glacier discharge contributions over the baseline period $Q_G(0)$ (mm) with mean observed specific discharge values $Q(0)$ (mm) for the large basins as shown in Fig. 1.

**Table 1: Comparison of baseline period mean specific subcatchment discharge $Q(0)$ with glacier runoff $Q_G(0)$ as provided by**

**(Rounce et al., 2023). In the absence of glaciation, $Q_G = 0$ for the current and future periods. Hence, in the Murghab-Harirud basin, there is not contribution to discharge from glaciers.**

| Basin | $Q(0)$ | $Q_G(0)$ | $Q(0) - Q_G(0)$ |
|---|---|---|---|
| **Amu Darya** | 382 mm | 33 mm | 349 mm |
| **Chu-Talas** | 329 mm | 53 mm | 276 mm |
| **Issyk-Kul** | 459 mm | 108 mm | 352 mm |
| **Murghab-Harirud** | 67 mm | 0 mm | 67 mm |
| **Syr Darya** | 415 mm | 23 mm | 391 mm |

Table 2 shows median specific glacier discharge values for the baseline period and the future periods p1 – p3 for all 5 large basins. Median values over all 4 SSP are shown. Compared to the other basins, these data suggest that peak water in the Amu

Darya only occurs in period p2. In the other basins peak water is projected to occur during period p1. This can be explained by much higher basin elevations and correspondingly larger glaciation at higher elevations.

**Table 2: Mean specific glacier discharge $Q_G$ in mm for the baseline and future target periods. Values averaged over the 4 SPP are shown. Data are aggregated and summarized from (Rounce et al., 2023). No data are available for Murghab-Harirud basin because**

**of an absence of glaciation there.**

| Basin | $Q_G(0)$ | $Q_G(1)$ | $Q_G(2)$ | $Q_G(3)$ |
|---|---|---|---|---|
| **Amu Darya** | 33 mm | 44 mm | 53 mm | 44 mm |



| | | | | |
|---|---|---|---|---|
| **Chu-Talas** | 53 mm | 66 mm | 56 mm | 44 mm |
| **Issyk-Kul** | 108 mm | 146 mm | 120 mm | 81 mm |
| **Murghab-Harirud** | 0 mm | 0 mm | 0 mm | 0 mm |
| **Syr Darya** | 23 mm | 32 mm | 32 mm | 22 mm |

## 3. Results and Discussion

### 3.1 Climate Change Analysis

First, we compute the relative changes in the climate variables, i.e., $dE_p/E_p(0)$, $d\alpha/\alpha(0)$, $d\lambda/\lambda(0)$ and $dw_0/w_0(0)$ as per Eq. (10). Remember that in this notation, the numbers in the parentheses are period identifiers. For example, $E_0(0)$ refers to the norm potential evaporation during the baseline period.

Figure 9 shows socio-economic pathway and target period dependent relative changes. In line with the expected temperature changes in the region, median potential evaporation values increase almost 20% relative to baseline values (Plate a). For all scenarios, the median relative increases in evaporation are at 3.91% for period p1, 7.48% for period p2, and 11.76% for period p3.

At the same time, we see a significant modulation of precipitation depths $\alpha$ and frequencies $\lambda$ over the entire projection period. Averaged over scenarios, median per period increases are +4.58%, +6.68% and +9.50% for p1, p2, and p3 respectively for $\alpha$ and -0.38%, -0.95%, and -1.76% for $\lambda$. On average, the projected decline in precipitation frequency is offset by a stronger increase in precipitation event depth, thus indicating a wetter future in Central Asia (+4.43%, +5.88%, and +8.49% of relative increases in median total precipitation values).

As for the absolute median delta changes in temperatures, projections are +1.33 °C, +2.44 °C, and +3.56 °C for the corresponding periods (not shown). Per our assumptions, these changes are proportional to the changes in the maximum water available for evaporation over the root zone. The large spread of the relative changes shown in Plate d in Fig. 9 stems from our simple model assumptions on how the dynamics of $w_0$ is determined by changes in norm temperatures for the corresponding target periods. The current model is agnostic to baseline starting values $w_0(0)$ and, in case of small initial values, can lead to a large percentage increase in such basins. Thus, for absolute median delta changes in $w_0$ we get +1.50%, 2.80%, and 4.13% correspondingly for the three future target periods.





## 3.2 Quantification of Climate Impacts

The distributions of the computed climate sensitivity indices as per Equations 11 – 14 are shown in Fig. 11. The $\sigma$ can be interpreted in the following way. For the vast majority of subcatchments, $\sigma_\alpha$ and $\sigma_\lambda$ are positive. This means that positive
relative changes of the corresponding climate variable translate into increases in relative discharge $dQ/Q(0)$, all else being equal. $\sigma_{E_p}$ and $\sigma_{w_0}$ are negative, this means that either relative increases in evaporation or in the maximum amount of soil water available for evaporation will translate into a decrease in discharge. The balancing effect of the climate sensitivities, their magnitudes and the projected relative changes will then determine the overall climate impacts in the PSM model.

Using the D66 $Z_r$ as representative rooting depth values, the scenario and target period specific relative changes in discharge as computed by Eq. (15) are shown in Fig. 12. The median change values do indicate significant changes across the region. However, there is a very large spread of values around the center. In some instances, the model results even suggest a more than 4-fold increase of relative discharge values (see positive outliers for SSP3-70 and SSP5-85, Period p3 in Fig. 12). These outliers are a clear indication that we need to be very careful when interpreting results at the individual subcatchment level.
Yet, one could also argue that a 300% increase in runoff in a very dry area will still be within the natural interannual variability in that catchment.

Nevertheless, taking into consideration the large uncertainties in this study, it is certainly more robust to quantify changes at the level of the large basins (Amu Darya, Chu Talas, Issyk-Kul, Murghab-Harirud, and Syr Darya, see Fig. 1) and to report
aggregate statistics for these to get a robust climate impact prediction. We will follow this approach subsequently.

For median value projections, clear trends in the PSM model projections are discernible when making comparison between scenarios and between periods. Averaged over scenarios, median per period increases are +5.58%, +8.50% and +12.53% for p1, p2, and p3 respectively. Correspondingly, averaged over the three future periods, median per scenario changes are +6.51%
(SSP1-26), +7.73% (SSP2-45), +9.03% (SSP3-70), and +10.24% (SSP5-85). The median model projections of the PSM model thus indicate increasing discharge over time on the one hand and increases in discharge in line with the forcing scenario on the other.

The individual contributions to the overall projected relative changes shown in Fig. 12 are depicted in Fig. 13 (right-hand side
4 terms in Eq. (15)). For all terms and all scenarios, significant outliers are visible. However, when we look at median values, we see corresponding trends and increasing impacts with time and with climate forcing which is a finding confirmed for the aggregate response as shown in Fig. 12.



Combining the results from the PSM model with the glacier dynamics, we get per period mean median specific discharge
values of p1: 326 mm (322 mm, 330 mm), p2: 328 mm (324 mm, 331 mm), and p3: 329 mm (325 mm, 331 mm). The values
in the parenthesis are minimum and maximum median values over the scenarios. These results suggest little to changes to
specific discharge for the future periods but represents nevertheless an increase over baseline values of 304 mm. However, this
view blends over significant region- and subcatchment-specific differences in the model projections. In this regard, Table 3
shows the statistics for the large basins with the corresponding climate scenario-related uncertainties. The median projected
discharge increases in all basins over time, except in the Syr Darya, where a reduction in median discharge values is project
from period p2 onwards (p2: 97% of baseline value, p3: 96% of baseline value). For the Murghab-Harirud basin, the PSM
model projects a significant increase of median specific discharge values of more than 100% in period p3 relative to the
baseline period value.

Hence, when we consider averaged over scenarios climate impacts, our model results together combined with modelled glacier
projects suggest an increase in available discharge throughout the mountain ranges in the Central Asia region, except for the
Syr Darya basin starting in the second half of the 20[th] century. The Central Asia region is not only heating up, very much in
line with much of the rest of the globe but at the same time, increases in discharge can potentially benefit the growing societies
there.


**Table 3: Median per period specific discharge values in mm computed with the PSM model climate sensitivity components and with added discharge from glaciers. Numbers in bold are median values averaged over the climate scenarios. The numbers in the parenthesis are minimum and maximum median values over the different scenarios and hence are an indication of scenario-related uncertainty.**

| Basin | Baseline | Period p1 | Period p2 | Period p3 |
|---|---|---|---|---|
| **Amu Darya** | 287 | **315** (307, 321) | **317** (313, 324) | **324** (310, 333) |
| **Chu-Talas** | 373 | **380** (380, 381) | **388** (381, 392) | **387** (368, 415) |
| **Issyk-Kul** | 525 | **571** (567, 573) | **588** (585, 594) | **600** (582, 617) |
| **Murghab-Harirud** | 55 | **86** (82, 92) | **91** (81, 103) | **108** (90, 115) |
| **Syr Darya** | 372 | **379** (366, 385) | **361** (351, 374) | **356** (346, 375) |


### 3.3 Sensitivity to Variations in Rooting Zone Depth $Z_r$

We have already discussed the significant differences between the individual soil rooting depth products previously (see for
example Fig. 6). How robust are the PSM model projections regarding changes in soil depth? To study the implication of such
changes, we systematically test different data sets of rooting zone depth and compute corresponding PSM model climate
impacts. Figure 14 shows the boxplot statistics of the different data for the basin storage index $\gamma$ used as a function of the
rooting zone depth.



Table 4 shows the resulting averaged over subcatchments and scenarios impacts. Period-specific ensemble means are +5.92%, +9.29%, and +14.31% respectively with corresponding standard errors of 0.08%, 0.17%, and 0.38%. As can easily be seen,
results are robust across the wide range of root zone depth values. This is encouraging as this robustness of the median projections in terms of varying soil properties lends certain credibility to the model projections and computed climate impacts.

**Table 4: Comparison of overall scenario-averaged median values of relative changes in discharge as computed by the PSM model. The D66 Fan $Z_r$ (fan_d66 data in top row) are the reference values used in this study.**

| Data Source | Period p1 | Period p2 | Period p3 |
|---|---|---|---|
| fan_d66 | +5.58% | +8.50% | +12.53% |
| fan_d80 | +5.87% | +8.92% | +13.17% |
| fan_d95 | +5.77% | +9.07% | +13.82% |
| fan_d95_yang_d99 | +5.71% | +9.04% | +13.98% |
| fan_d99 | +6.06% | +9.43% | +14.66% |
| fan_stocker_yang_d99 | +6.17% | +9.84% | +15.51% |
| s0_stocker | +6.06% | +9.71% | +15.08% |
| stocker_d99 | +6.34% | +10.12% | +16.05% |
| yang_d99 | +5.71% | +8.98% | +13.95% |


## 4. Discussion and Conclusions

For the study of climate impacts in the high mountain Central Asia region, we have used a simple, parsimonious conceptual model of soil moisture dynamics to account for essential components of the soil water balance at the catchment level. The model includes soil water storage, threshold-triggered runoff and deep infiltration, and water storage dependent evaporation.
State variables and fluxes are driven by intermittent stochastic rainfall whose location-dependent variability is characterized in terms of precipitation frequency and depth.

The PSM model has limitations. First, land ice is not represented in the model. This means, for example, that dynamics and implications of destorage due to glacier melt cannot be studied with it which is why we use available data about the future
mass balance of glaciers in the region to study percentage contributions to discharge from glacier melt over the course of the 21[st] century. Second, neither seasonal nor permanent snow storage is accounted for in the model. Research, however, has shown that the scatter of real-world data around Budyko's curve is also influenced by catchment snow ratios (Zhang et al., 2015), apart from the other controls such as vegetation, the seasonality characteristics of precipitation, soil properties (as implemented in the PSM model) and topographic factors (Greve et al., 2016). Third, in our formulation of climate sensitivity,



we only indirectly account for future permafrost dynamics via the $T$ and $w_0$ linear relationship while computing $dw_0/w_0$ climate sensitivities. Permafrost thawing and additional contributions to discharge are not taken into considerations here.

While being computationally extremely efficient, the analytical solution of the PSM model is limited to steady-state conditions. The model helps us to understand total water availability and inter-period changes therein. It assumes instantaneous shifts

between target climate periods and specific annual norm discharge regimes that apply uniformly throughout these periods. At the same time, impacts on discharge seasonality cannot be studied. For example, with increasing warming and the shift of discharge regimes from nival-glacial to pluvial-nival, cold season discharge between October and March will increase and, with that, the potential for hydropower production during the times where electricity demand for heating is high.

Second, as Siegfried et al. (2012) has shown focusing on the Syr Darya river basin, the mismatch between water availability during the warm season and crop water demand will increase. With a rebalancing of cold to warm season discharge ratios to higher values and strongly increasing evaporative water demands in the downstream in irrigated agriculture (for irrigation and crop cooling alike), there will be an increasing water gap emerging that can only be managed in selected basins with the help of sufficient-volume man-made storage. Under the assumption that agricultural sector in the countries of the region continue

to be important for food self-sufficiency and the production of cash crops as well as for livelihoods, a boom in the construction of reservoirs will likely occur. These reservoirs not only can offset the loss of glacier water storage but, when equipped with hydropower, can contribute to a green energy transition of the region. However, care must be taken that habitat fragmentation in the mountain rivers does not translate into adverse ecological outcomes under such scenario.

The careful analysis of climate projections over 221 subcatchments in the region reveals important changes in precipitation frequency and event depths. Both show patterns across the mountain ranges as shown in Fig. 15. The developments shown in this Figure, exemplified by the display of the period p3 conditions under SSP5-85 indicates places where both frequency and depth increase significantly relative to baseline values. This not only indicates the potential for a wetter future in these subcatchments but also points to the importance of gaining a better understanding of the evolution of climate extremes in

Central Asia.

Whereas we gain a good understanding of the development of an average water availability in the region with the present study, more emphasis needs to be put in the future to the study of local hydrologically relevant extremes. This includes both wet and dry extremes that can equally have very adverse impacts on relevant segments of society.


In line with the observed large-scale regional trends in discharge as shown in Fig. 1, our modeling results indicate a wetter future as compared to the baseline period. Care must be taking not to draw the wrong conclusioqns. The setting remains to be



one of extraordinary fragility and care must be taken to make resource use more efficient and more effective, notwithstanding the results presented here.


To summarize, we would like to emphasize that the model results need to be interpreted with care. First, we are using an extraordinarily simple model to study complex real-world processes. The analytical solution of the model shows the importance of climate and soil characteristics determining overall climate sensitivities. Marti et al. (2023) shows that the best high-resolution climatology products over the region suffer from built-in problems that stem from the fact that the observation

network greatly decayed during the 1990ies after the political transition. The global products available in relation to soil characteristics are highly variable and there is little confidence that the derived subcatchment averaged values correspond to actual effective values.

There are many ways to extend the current model. One obvious way to develop a more realistic representation of the

hydrological situation in high mountain Central Asia would be trying to incorporate snow in the model. Berghuijs et al. (2014) have used empirical evidence using data from the contiguous United States that a decreasing snow fraction in precipitation (proportion of solid precipitation as a fraction of total precipitation) is associated with lower streamflow at the annual and multi-annual time scales. This effect then would counter the increase in discharge by the PSM model used here. Under any circumstances, analytical solutions of the PSM Model might no longer be available with such model extensions.

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



**Appendices**

**Appendix A Derivation of Climate Sensitivities**

The Porporato simplified stochastic soil moisture dynamics model (PSM) is defined as

$$\frac{E}{P} = f(\phi, \gamma)$$

where $E$ is actual norm evaporation and $P$ is norm precipitation and

$$f(\phi, \gamma) = 1 - \frac{\phi \gamma^{\frac{\gamma}{\phi}-1} e^{-\gamma}}{\Gamma\left(\frac{\gamma}{\phi}\right) - \Gamma\left(\frac{\gamma}{\phi}, \gamma\right)}$$

Under steady state conditions, we have $E/P = 1 - Q/P$, hence


$$\frac{Q}{P} = g(\phi, \gamma)$$

where

$$g(\phi, \gamma) = \frac{\phi \gamma^{\frac{\gamma}{\phi}-1} e^{-\gamma}}{\Gamma\left(\frac{\gamma}{\phi}\right) - \Gamma\left(\frac{\gamma}{\phi}, \gamma\right)}$$

For the computation of the climate sensitivity indices, we compute the following total derivative $D[]$ as

$$D[Q = \alpha \, \lambda \, g(\frac{E_p}{\alpha \lambda}, \frac{w_0}{\alpha})]$$


In this formulation, precipitation $P$ and the aridity and storage indices $\phi$ and $\gamma$ have been replaced by their constituting terms $\alpha$ (precipitation depth), $\lambda$ (precipitation frequency), $E_p$ (potential evaporation), and $w_0$ (max. soil water available for evaporation). It can easily be shown that this evaluates to

$$dQ = g^{(1,0)}\left(\frac{E_p}{\alpha\lambda}, \frac{w_0}{\alpha}\right) dE_p + \lambda g^{(0,1)}\left(\frac{E_p}{\alpha\lambda}, \frac{w_0}{\alpha}\right) dw_0 + \lambda \, g\left(\frac{E_p}{\alpha\lambda}, \frac{w_0}{\alpha}\right) d\alpha - \frac{E_p}{\alpha} g^{(1,0)}\left(\frac{E_p}{\alpha\lambda}, \frac{w_0}{\alpha}\right) d\alpha -$$





$$\frac{\lambda w_0}{\alpha} g^{(0,1)} \left(\frac{E_p}{\alpha\lambda}, \frac{w_0}{\alpha}\right) d\alpha + \alpha\, g\left(\frac{E_p}{\alpha\lambda}, \frac{w_0}{\alpha}\right) d\lambda - \frac{E_p}{\lambda} g^{(1,0)}\left(\frac{E_p}{\alpha\lambda}, \frac{w_0}{\alpha}\right) d\lambda$$

where the superscripts of the $g(.,.)$ function denote the corresponding partial differential. We can rearrange and collect the terms and express the changes in relative terms. As a result, we get


$$\frac{dQ}{Q} = \left[\frac{E_p}{Q} g^{(1,0)}\left(\frac{E_p}{\alpha\lambda}, \frac{w_0}{\alpha}\right)\right] \frac{dE_p}{E_p} + \left[1 - \frac{E_p}{Q} g^{(1,0)}\left(\frac{E_p}{\alpha\lambda}, \frac{w_0}{\alpha}\right) - \frac{\lambda w_0}{Q} g^{(0,1)}\left(\frac{E_p}{\alpha\lambda}, \frac{w_0}{\alpha}\right)\right] \frac{d\alpha}{\alpha} +$$

$$\left[1 - \frac{E_p}{Q} g^{(1,0)}\left(\frac{E_p}{\alpha\lambda}, \frac{w_0}{\alpha}\right)\right] \frac{d\lambda}{\lambda} + \left[\frac{\lambda w_0}{Q} g^{(0,1)}\left(\frac{E_p}{\alpha\lambda}, \frac{w_0}{\alpha}\right)\right] \frac{dw_0}{w_0}$$

Note that we exploited that here the fact that $Q/P = g(\phi, \gamma)$. The terms in the squared brackets are the corresponding climate
sensitivity coefficients. We can introduce the following simplifications

$$\sigma_{E_p} = \frac{E_p}{Q} g^{(1,0)}\left(\frac{E_p}{\alpha\lambda}, \frac{w_0}{\alpha}\right)$$

$$\sigma_{w_0} = \frac{\lambda w_0}{Q} g^{(0,1)}\left(\frac{E_p}{\alpha\lambda}, \frac{w_0}{\alpha}\right)$$

and thus write the relative changes in a compact form as

$$\frac{dQ}{Q} = \sigma_{E_p} \frac{dE_p}{E_p} + \left(1 - \sigma_{E_p} - \sigma_{w_0}\right) \frac{d\alpha}{\alpha} + \left(1 - \sigma_{E_p}\right) \frac{d\lambda}{\lambda} + \sigma_{w_0} \frac{dw_0}{w_0}$$

$\sigma_{E_p}$ and $\sigma_{w_0}$ can be computed as follows. For $\sigma_{E_p}$, we get


$$\sigma_{E_p} = \frac{E_p}{Q} g(\phi, \gamma) \left(\frac{1}{\phi} - \frac{\gamma^{\frac{\gamma}{\phi}-1}}{\Gamma\left(\frac{\gamma}{\phi}\right) - \Gamma\left(\frac{\gamma}{\phi}, \gamma\right)} {}_2F_2\left(\frac{\gamma}{\phi}, \frac{\gamma}{\phi}; \frac{\gamma}{\phi}+1, \frac{\gamma}{\phi}+1; -\gamma\right)\right)$$

which is

$$\sigma_{E_p} = \phi\left(\frac{1}{\phi} - \frac{e^\gamma}{\phi} g(\phi, \gamma)\, {}_2F_2\left(\frac{\gamma}{\phi}, \frac{\gamma}{\phi}; \frac{\gamma}{\phi}+1, \frac{\gamma}{\phi}+1; -\gamma\right)\right)$$



and finally gets

$$\sigma_{E_p} = 1 - e^\gamma g(\phi, \gamma) \, {}_2F_2\left(\frac{\gamma}{\phi}, \frac{\gamma}{\phi}; \frac{\gamma}{\phi} + 1, \frac{\gamma}{\phi} + 1; -\gamma\right)$$


With the definition of $\chi = e^\gamma g(\phi, \gamma) {}_2F_2\left(\frac{\gamma}{\phi}, \frac{\gamma}{\phi}; \frac{\gamma}{\phi} + 1, \frac{\gamma}{\phi} + 1; -\gamma\right)$, we can retrieve Eq. (11) from Section 2.3 above where $\sigma_{E_p} = 1 - \chi$. The relative changes in discharge thus become

$$\frac{dQ}{Q} = (1 - \chi)\frac{dE_p}{E_p} + \left(\chi - \sigma_{w_0}\right)\frac{d\alpha}{\alpha} + \chi\frac{d\lambda}{\lambda} + \sigma_{w_0}\frac{dw_0}{w_0}$$


For $\sigma_{w_0}$, we get correspondingly

$$\sigma_{w_0} = \frac{\lambda w_0}{\gamma^2 Q} g(\phi, \gamma)\left(\frac{e^{-\gamma}\gamma^{\frac{\gamma}{\phi}}}{\Gamma\left(\frac{\gamma}{\phi}\right) - \Gamma\left(\frac{\gamma}{\phi}, \gamma\right)}\left(e^\gamma \phi \, {}_2F_2\left(\frac{\gamma}{\phi}, \frac{\gamma}{\phi}; \frac{\gamma}{\phi} + 1, \frac{\gamma}{\phi} + 1; -\gamma\right) - \gamma\right) + \frac{\gamma(\gamma - \phi(\gamma + 1))}{\phi}\right)$$

which is

$$\sigma_{w_0} = \frac{1}{\gamma}\left(\gamma\frac{e^{-\gamma}\gamma^{\frac{\gamma}{\phi}-1}}{\Gamma\left(\frac{\gamma}{\phi}\right) - \Gamma\left(\frac{\gamma}{\phi}, \gamma\right)}\left(e^\gamma \phi \, {}_2F_2\left(\frac{\gamma}{\phi}, \frac{\gamma}{\phi}; \frac{\gamma}{\phi} + 1, \frac{\gamma}{\phi} + 1; -\gamma\right) - \gamma\right) + \frac{\gamma(\gamma - \phi(\gamma + 1))}{\phi}\right)$$

Further simplifying, we obtain


$$\sigma_{w_0} = g(\phi, \gamma)\left(e^\gamma \, {}_2F_2\left(\frac{\gamma}{\phi}, \frac{\gamma}{\phi}; \frac{\gamma}{\phi} + 1, \frac{\gamma}{\phi} + 1; -\gamma\right) - \frac{\gamma}{\phi}\right) + \frac{\gamma - \phi(\gamma + 1)}{\phi}$$

and

$$\sigma_{w_0} = \chi - \frac{\gamma}{\phi}g(\phi, \gamma) + \frac{\gamma - \phi(\gamma + 1)}{\phi}$$

or



$$\sigma_{w_0} = \chi + \frac{\gamma}{\phi}\left(1 - g(\phi, \gamma)\right) - (\gamma + 1)$$


Hence, we get for the final relative changes in discharge

$$\frac{dQ}{Q} = (1 - \chi)\frac{dE_p}{E_p} + \left(-\frac{\gamma}{\phi}\left(1 - g(\phi, \gamma)\right) + (\gamma + 1)\right)\frac{d\alpha}{\alpha} + \chi\frac{d\lambda}{\lambda} + \left(\chi + \frac{\gamma}{\phi}\left(1 - g(\phi, \gamma)\right) - (\gamma + 1)\right)\frac{dw_0}{w_0}$$

where the expressions in the square brackets correspond to the ones shown in Eq. (11) – (14).

**Appendix B Soil-Related Parameters**

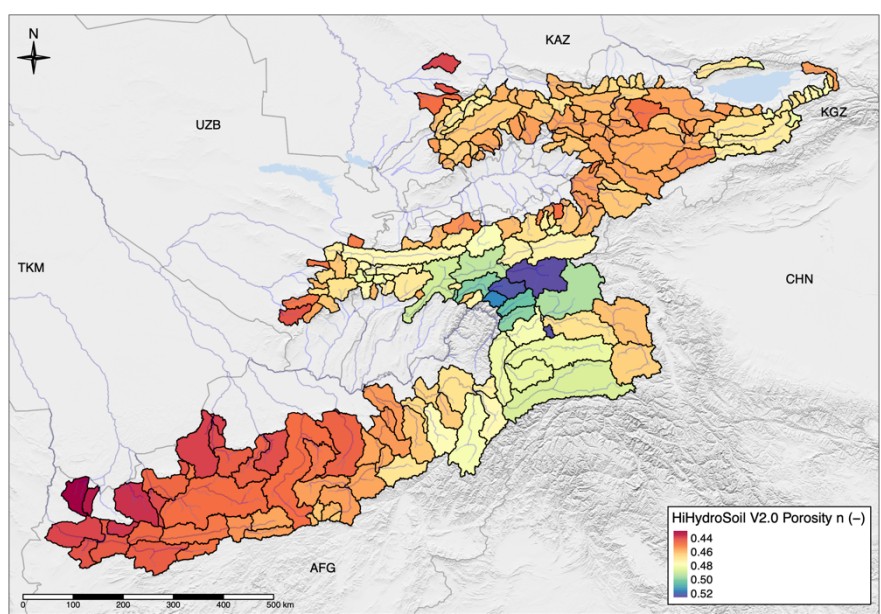

**Figure B1: Subcatchment averaged HiHydroSoil V2.0 porosity n values are shown (WCsat dataset). We assume that these values**
**are corresponding to subcatchment-averaged saturation excess values $s_1$ (-). See (Simons et al., 2020) for more information.**



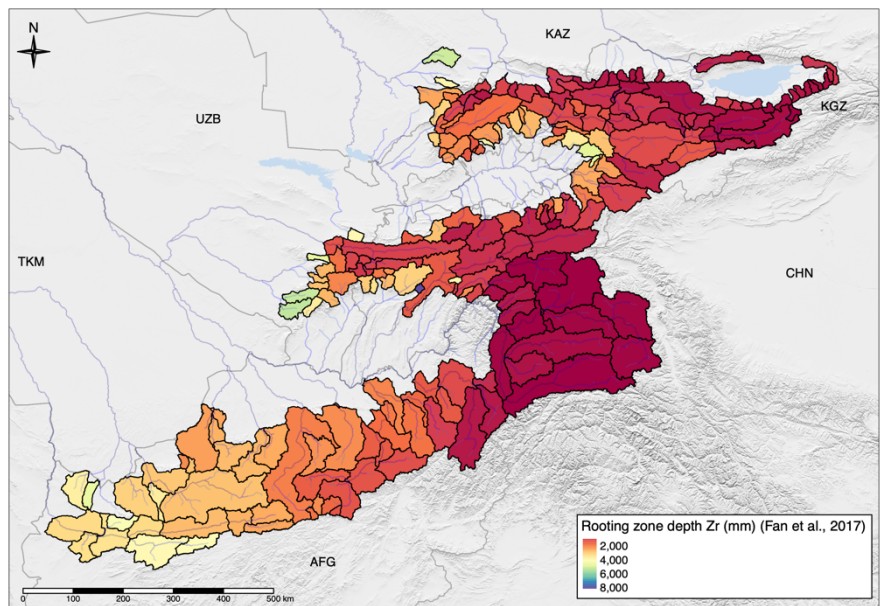

**Figure B2: Subcatchment averaged rooting zone depth values from (Fan et al., 2017).**

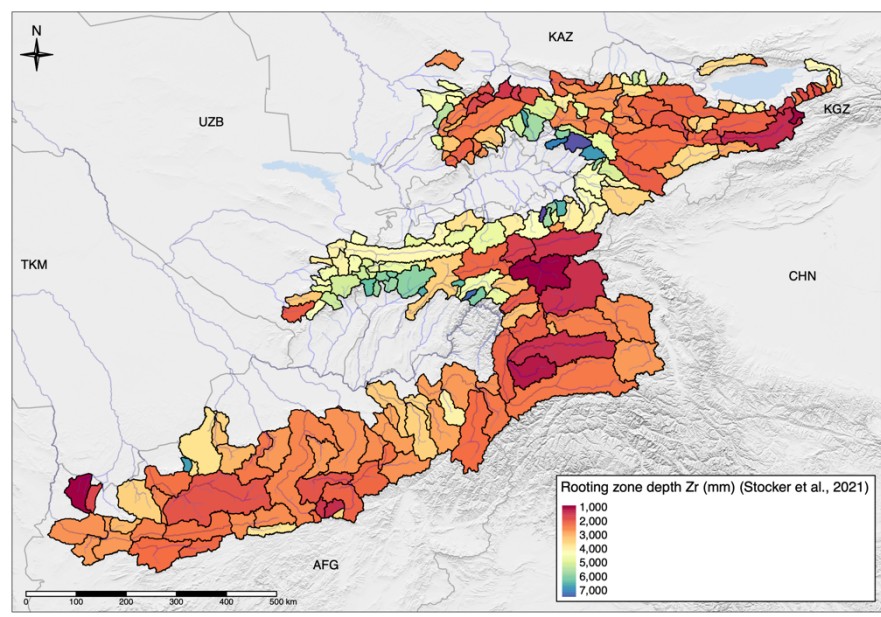


**Figure B3: Subcatchment averaged rooting zone depth values from (Stocker et al., 2021).**



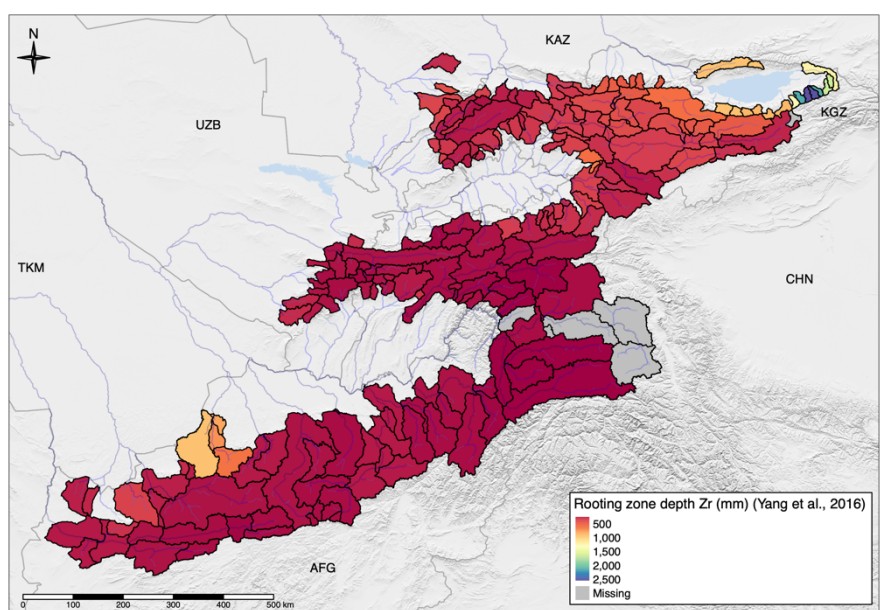

**Figure B4: Subcatchment averaged rooting zone depth values from (Yang et al., 2016).**


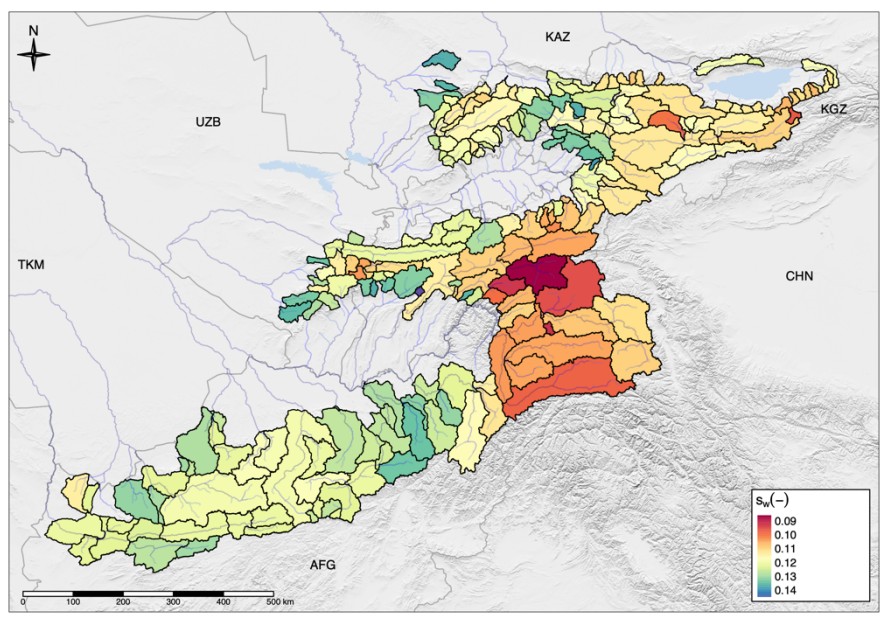

**Figure B5: Subcatchment averaged values of the wilting point $s_w$(WCpF4.2) dataset from (Simons et al., 2020).**



**Appendix C Baseline Rooting Depth and Land Cover Class Mapping**

For each subcatchment studied, data from the Copernicus 100 m 2019 land cover product is available in terms of area covered by each land cover class (Buchhorn et al., 2019). The complete list of land cover classes occurring in the presented data set include: 20 (Shrubs), 30 (herbaceous vegetation), 40 (cultivated and managed vegetation/agriculture), 50 (urban/built up), 60 (bare/sparse vegetation), 70 (snow and ice), 80 (open water), 90 (herbaceous wetland), 100 (moss and lichen), 111 (closed forest, evergreen needle leaf), 112 (closed forest, evergreen broad leaf), 113 (classified as closed forest, deciduous needle leaf),

114 (classified as closed forest, deciduous broad leaf), 115 (closed forest, mixed), 116 (closed forest, unknown), 121 (open forest, evergreen needle leaf), 122 (open forest, evergreen needle leaf), 123 (open forest, deciduous needle leaf), 124 (open forest, deciduous broad leaf), 125 (open forest, mixed), and 126 (open forest, unknown).

The data by (Fan et al., 2017) reports maximum (modeled) rooting zone depth globally on a 30-arcsecond global grid. We

assume that these values correspond to the D99 depth values above which 99 % of the root biomass is located (Hauser et al., 2022). For the computation of a subcatchment-averaged mean rooting depth $Z_r(0)$, we compute the area weighted D99/D66 correction factors and then multiply these with the sub-catchment averaged values by (Fan et al., 2017). The detailed information is shown in Table A1.

**Table A1: Correspondence mapping between D99 values (Hauser et al., 2022), GLC Global Class (https://forobs.jrc.ec.europa.eu/products/glc2000/legend.php) and the Copernicus land cover class (Buchhorn et al., 2019). NA denotes no values. D99 values are in meters. LCC is land cover class, WT is water table. D66 values are computed with information from Table S1 in (Hauser et al., 2022)**

| GLC Global Class (according to LCCS terminology) | Copernicus LCC | D99 (m) | D66 (m) | D99/D66 (-) |
|---|---|---|---|---|
| Tree Cover, broadleaved, evergreen | lc_112, lc_122 | 1.41 | 0.82 | 15.0 |
| Tree Cover, broadleaved, deciduous, closed | lc_114 | 2.17 | 1.265 | 25.5 |
| Tree Cover, broadleaved, deciduous, open | lc_124 | 1.13 | 0.665 | 12.0 |
| Tree Cover, needle-leaved, evergreen | lc_111, lc_121 | 1.87 | 1.095 | 28.8 |
| Tree Cover, needle-leaved, deciduous | lc_113, lc_123 | 0.85 | 0.497 | 217.9 |
| Tree Cover, mixed leaf type | lc_115, lc_125 | 1.3 | 0.76 | 18.1 |
| Tree Cover, regularly flooded, fresh water (& brackish) | NA | NA | NA | NA |
| Tree Cover, regularly flooded, saline water, (daily WT var) | NA | NA | NA | NA |
| Mosaic: Tree cover / Other natural vegetation | lc_116, lc_126 | 2.4 | 1.43 | 9.7 |
| Tree Cover, burnt | NA | 2.4 | 1.43 | 9.7 |
| Shrub Cover, closed-open, evergreen | lc_20 | NA | NA | NA |
| Shrub Cover, closed-open, deciduous | lc_20 | 1.9 | 1.11 | 23.5 |
| Herbaceous Cover, closed-open | lc_30 | 1.58 | 0.92 | 87.8 |
| Sparse Herbaceous or sparse Shrub Cover | lc_60 | 3.1 | 1.83 | 15.7 |
| Regularly flooded Shrub and/or Herbaceous Cover | NA | NA | NA | NA |
| Cultivated and managed areas | lc_40 | 1.22 | 0.714 | 34.4 |



| | | | | |
|---|---|---|---|---|
| Mosaic: Cropland / Tree Cover / Other natural vegetation | lc_40 | 1.22 | 0.714 | 34.4 |
| Mosaic: Cropland / Shrub or Grass Cover | lc_40 | 1.22 | 0.714 | 34.4 |
| Bare Areas | lc_60 | 1.67 | 0.975 | 87.9 |
| Water Bodies | lc_80 | 0 | 0 | NA |
| Snow and Ice | lc_70 | 0.852 | 0 | NA |
| Artificial surfaces and associated areas | lc_50 | 1.35 | 0.79 | 61.4 |




## Figures



**Figure 1: Overview of the study region. 221 subcatchments shared among the 5 large river basins are shown. Large basins are color-coded (see legend). Long-term norm discharge data from these stations were collected and made available for this study/project (Marti et al., 2023). Gauge locations are indicated by dots. Blue colouring of these indicates increasing mean annual discharge in existing observation records when measured by the Sen's slope. Red colour is negative discharge trend and grey are locations where no time series is available for analysis. The hillshade layer is SRTM topography (NASA JPL, 2013b). Permanent water bodies were taken from the global HydroLakes Database (Messager et al., 2016). River shapes are from the WMOBB River Network data (GRDC, Koblenz, Germany: Federal Institute of Hydrology (BfG)., 2020). Glaciers are indicated by white polygons and are taken from the Randolph Glacier Inventory Version 6 (RGI Consortium, 2017).**



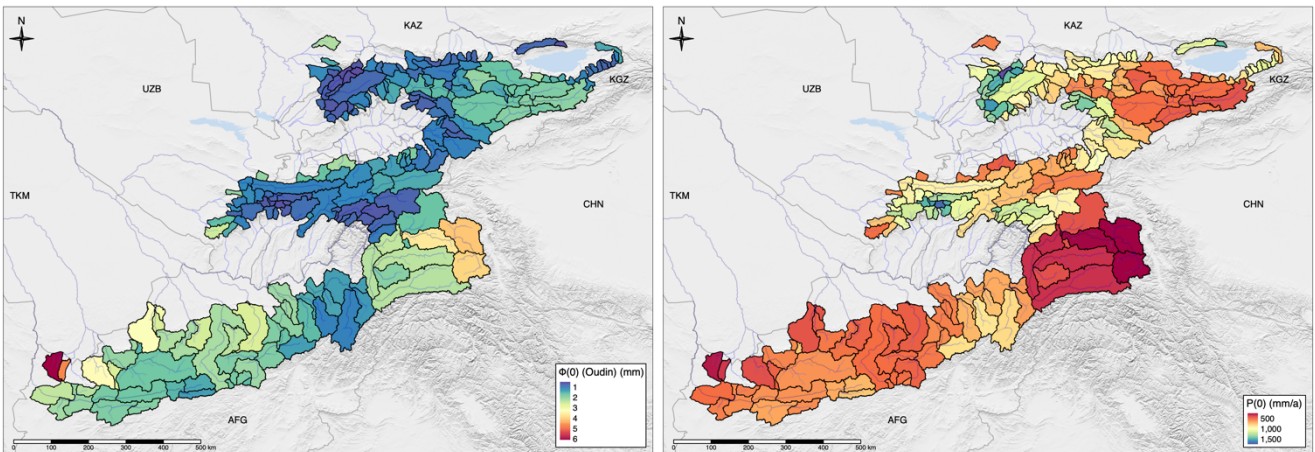

**Figure 2: Left plate: Mean aridity index $\phi(0) = E_0(0)/P(0)$ averaged over subcatchments for the baseline period (0). $E_0(0)$ has been calculated as in Eq. (16) and then by bias correcting with CHELSA V21 Penman values. Right plate: Baseline norm annual precipitation values from the CHELSA V21 climatology.**

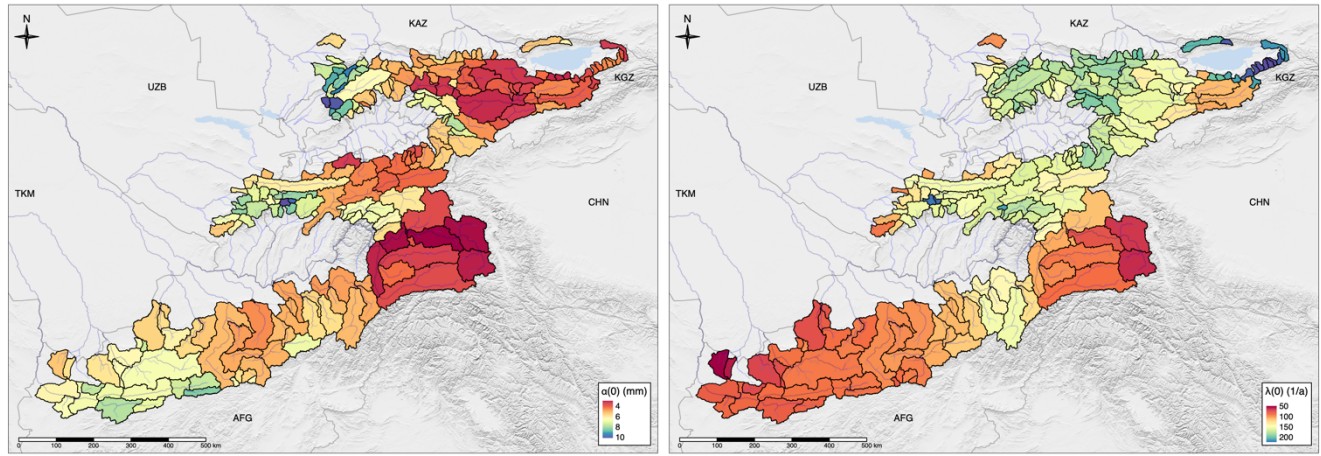

**Figure 3: Left plate shows the mean depth of precipitation events (mm) for each subcatchment for the baseline period. Right plate shows mean precipitation frequency values for each subcatchment for the baseline period ($a^{-1}$). In both instances, data are derived from daily high resolution precipitation CHELSA V21 climatologies.**





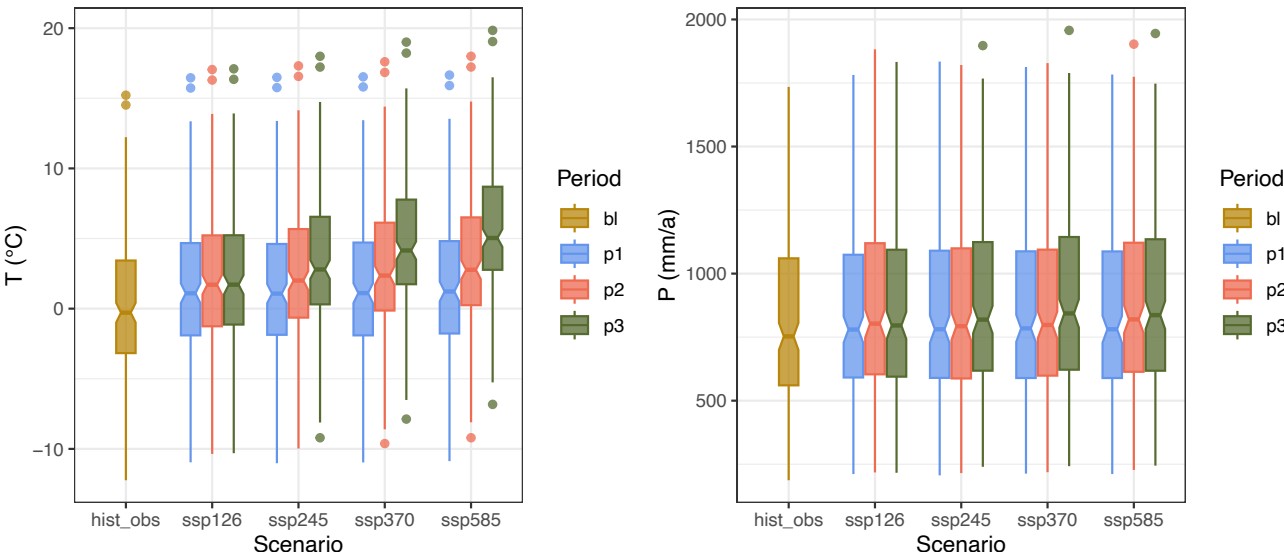

**Figure 4: Left plate shows the scenario dependent GCM averaged distributions of surface temperatures averaged over the subcatchments for the individual periods. The baseline period from 1979 until 2011 is denoted with bl. The left plate shows the corresponding figures in relation to the developments of annual norm precipitation values over all 221 subcatchments and for the individual periods. The boxplots visualize median values (thick black horizontal lines inside notches), first and third quartiles via the hinges. The upper whisker extends from the hinge to the largest value no further than 1.5 times the interquartile range (IQR, distance between first and third quartile) from the hinge. The lower whisker extends from the hinge to the smallest value at most 1.5 * IQR of the hinge. Data beyond the end of the whiskers are outliers that are plotted individually.**



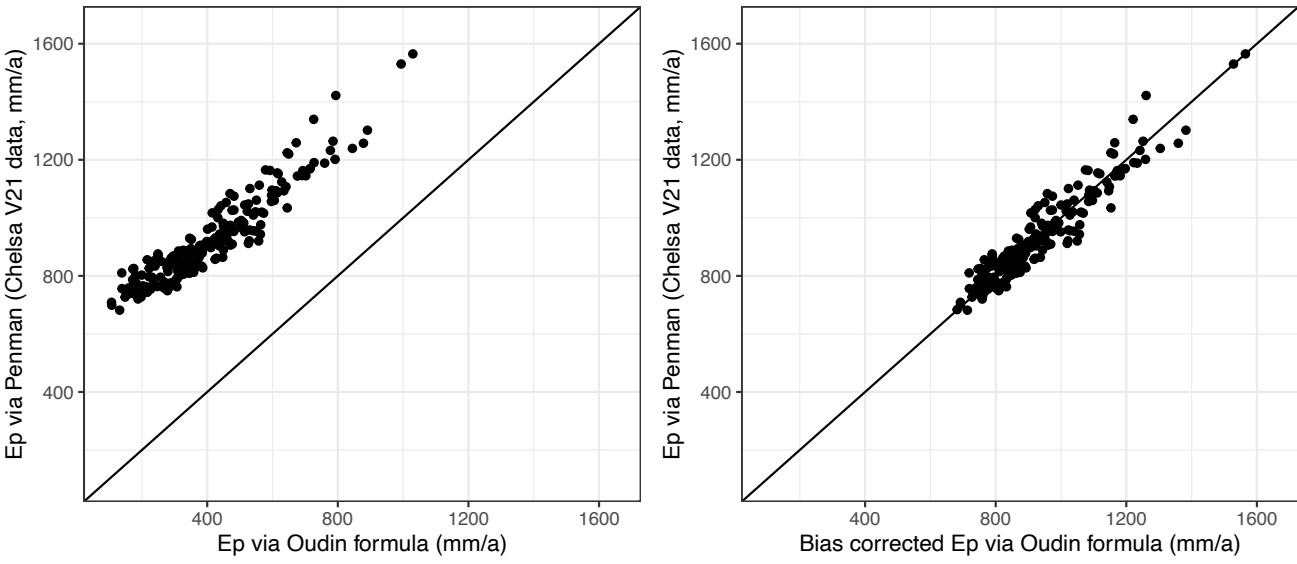



**Figure 5: The left plate compares average subcatchment $E_p$ as computed using Eq. (15) with $k_1 = 100$ and $k_2 = 5$ with CHELSA V21 Penman $E_p$. The former leads to significantly underestimated $E_p$. The right plate shows the resulting baseline $E_p$ values after bias correction.**


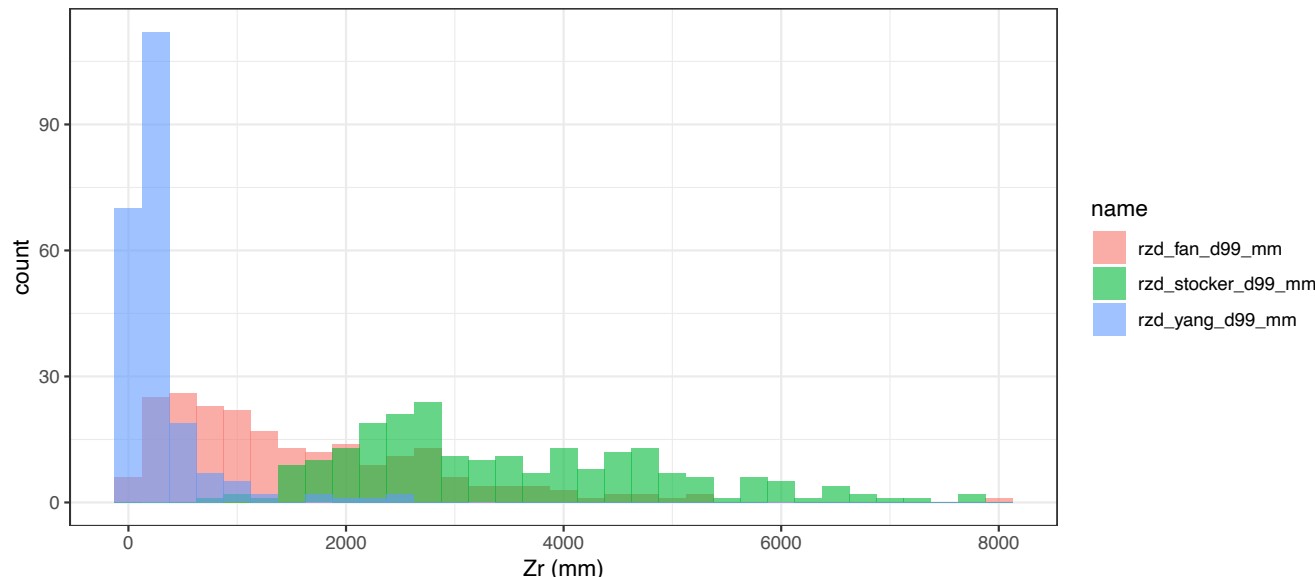

**Figure 6: Comparison of root zone depth $Z_r$ (mm) investigated (Fan et al., 2017; Stocker et al., 2021; Yang et al., 2016). The histograms show the distribution of values for the 221 catchments in the region for each data set. A large spread of the distribution of values is evident between the different available products.**


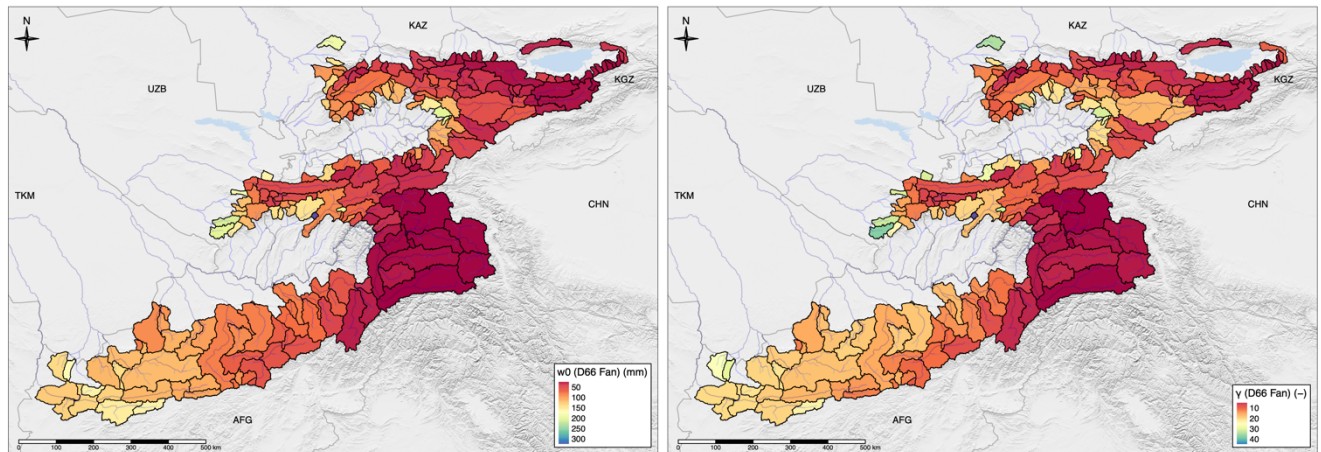

**Figure 7: Left plate: Maximum soil water storage for evaporation $w_0$ averaged over subcatchments computed using the rescaled (D95) soil rooting depth as reported by (Fan et al., 2017). Right plate: Corresponding subcatchment specific storage index $\gamma$ is shown.**





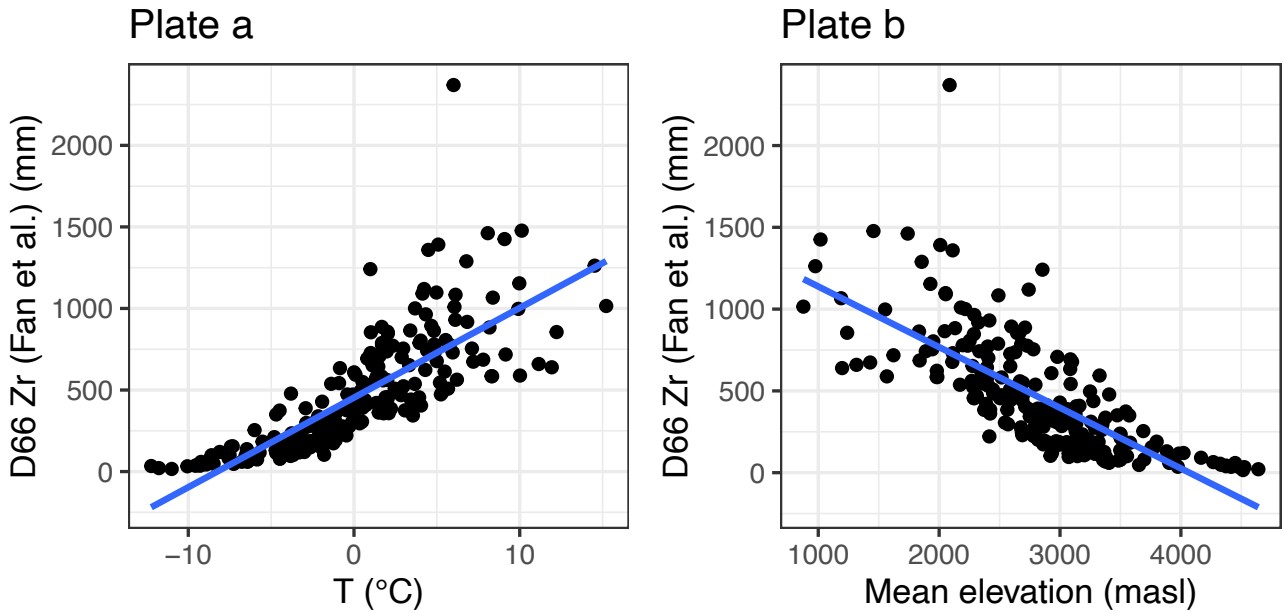


**Figure 8: Dependency of D66 $Z_r$ values on baseline norm temperature T (left Plate a) and mean basin elevation as derived from SRTM data (right Plate b). The blue line shows the best fit linear model with a slope of 115.7 mm/deg.C for the temperature dependency and -78 mm / 100 meters for the elevation dependency. See text for more information.**

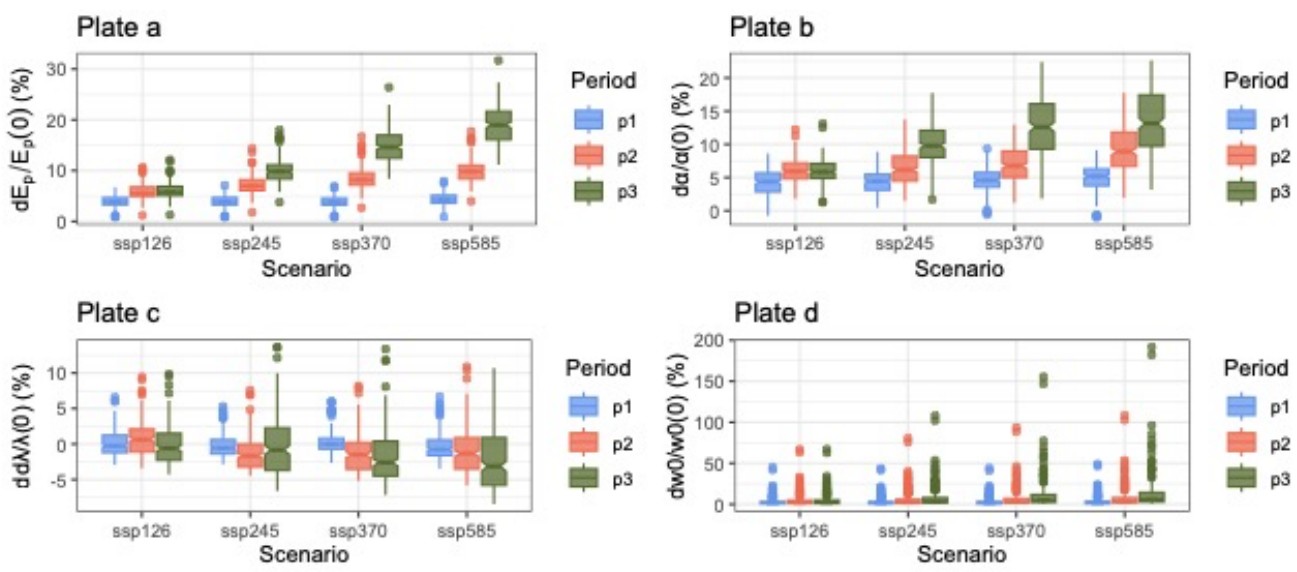


**Figure 9: Plate a: relative changes (in %, relative to baseline) in potential evaporation $E_p$ for different scenarios and target periods. Plate b: relative changes (in %, relative to baseline) in precipitation depth α for different scenarios and target periods. Plate c:**



relative changes (in %, relative to baseline) in precipitation frequency λ for different scenarios and target periods. Plate d: relative changes (in %, relative to baseline) in $w_0$ for different scenarios and target periods.


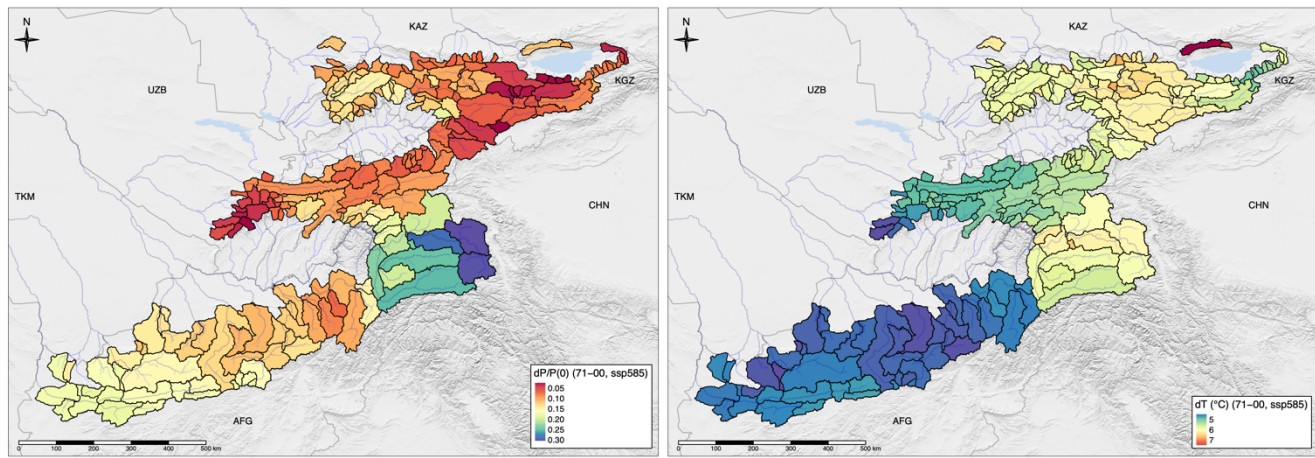

**Figure 10: Left plate: Map of projected relative increases in precipitation values (-) for the extreme SSP5-85 scenario and period p3 (2071-2100). Interestingly, the southern part of the domain including Afghanistan and the Pamirs are expected to see the highest relative precipitation increases. Right plate: absolute increases in projected mean period temperatures relative to baseline value.**
**While the whole region is expected to undergo significant warming, the most extreme developments are expected to take place in the high mountain parts of the Pamirs and the Tien Shan.**

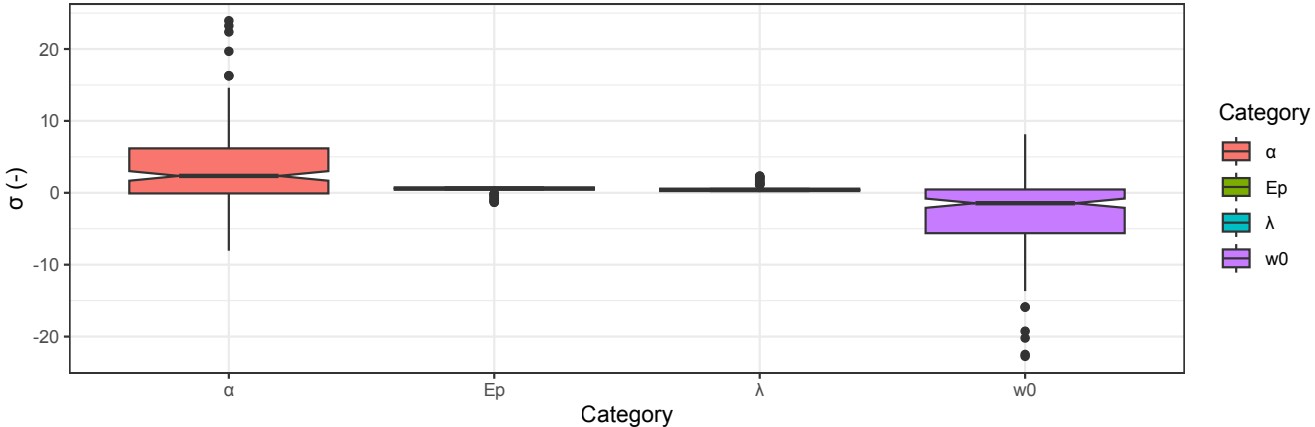

**Figure 11: Distributions of the four analytically calculated climate sensitivity indices are shown (see Eq. (11) – (14)). D66 $Z_r$ are used**
**for computation.**



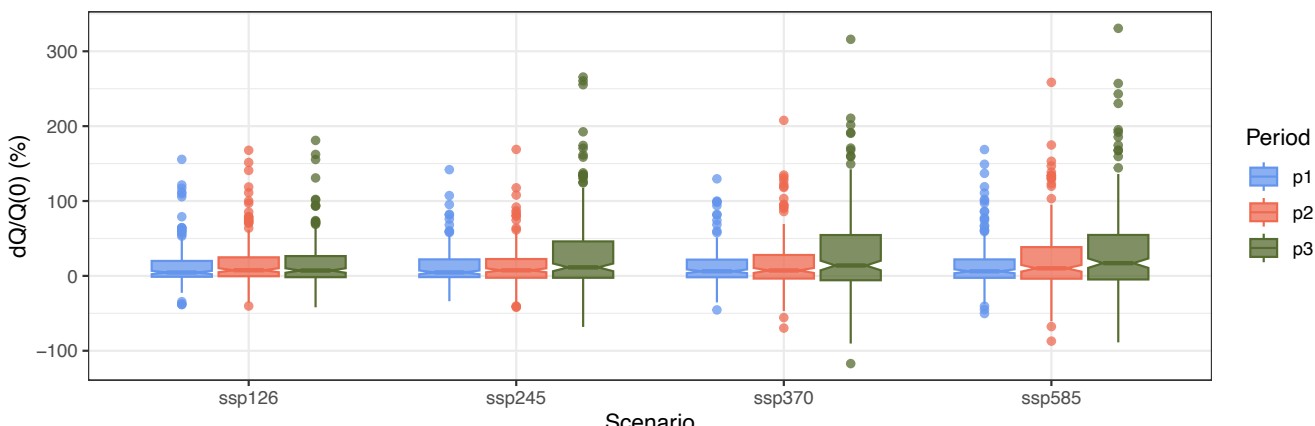

**Figure 12: Using the D66 $Z_r$ rooting depth data for computation, the boxplot statistics of the scenario and period-dependent relative changes to discharge are shown. Dots are considered outliers.**


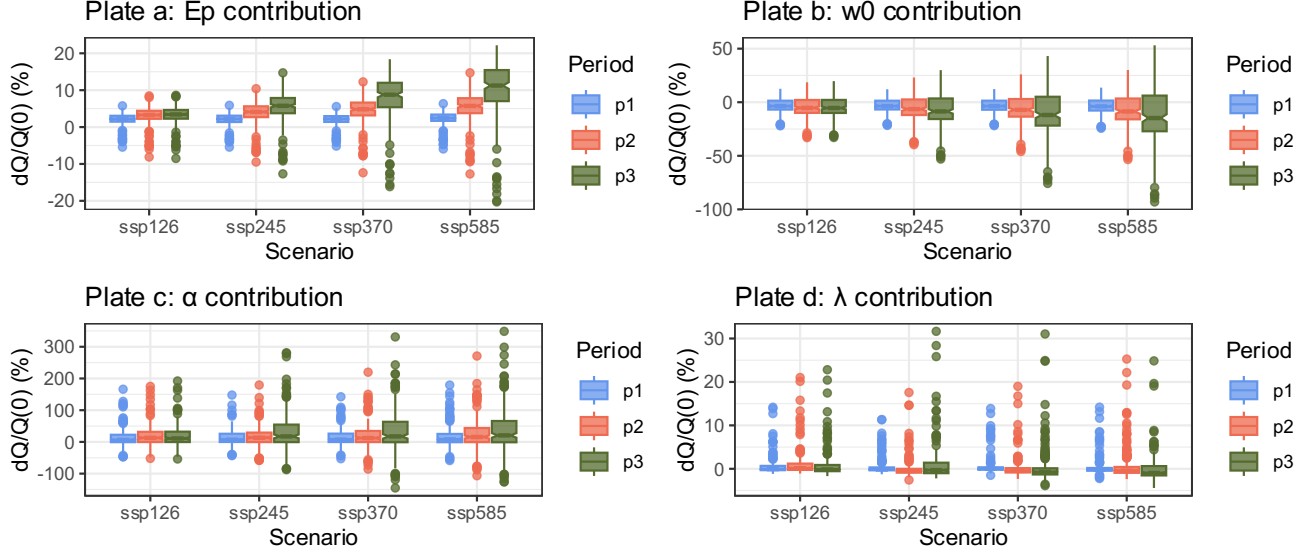

**Figure 13: Component contributions to the overall relative change in discharge using the D66 $Z_r$ rooting depth data. Significant outliers are visible in all plates. The corresponding median values are regarded as measures of robust projections. Dots are outliers.**




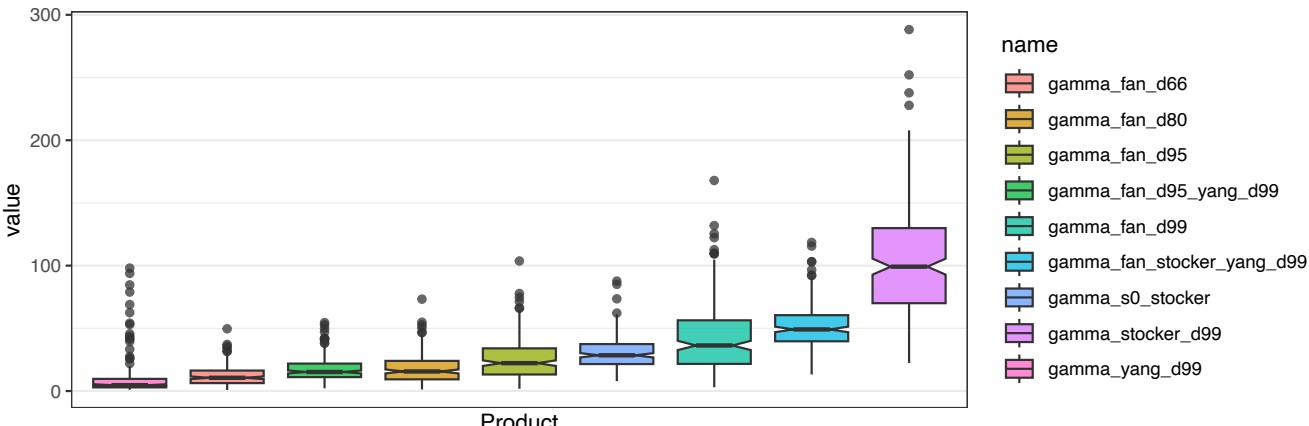

**Figure 14: Statistics of the basin storage indices γ computed with different source data of root zone depth $Z_r$. In ascending order from left to right, these are: 1) D99 Yang (Yang et al., 2016), 2) D66 Fan (Hu et al., 2018), 3) mean between D95 Fan $Z_r$ and D99 Yang $Z_r$, 4) D80 Fan, 5) D95 Fan, 6) $S_0$ by Stocker (Stocker et al., 2021), 7) D99 Fan, 8) mean between D99 Fan $Z_r$, D99 Stocker $Z_r$, and D99 Yang, and 9) D99 Stocker.**

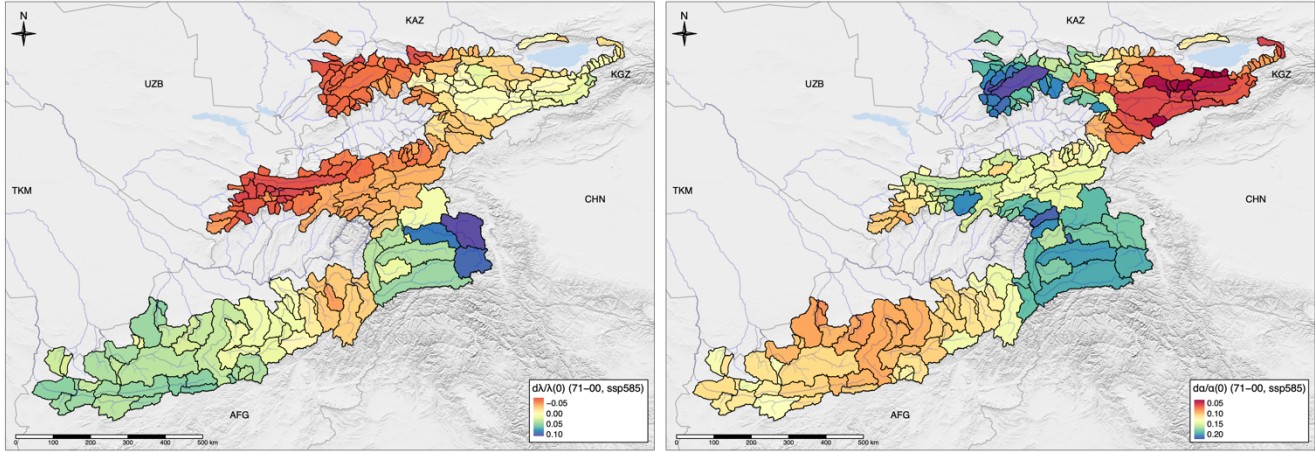

**Figure 15: The left plate shows the geographic distribution of precipitation frequency across the domain for SSP5-85 and the target period p3 (2071-2100). The right plate shows corresponding changes in precipitation depths. Compare also with Figure 3 and the left plate of Figure 7.**