# Peer review of "Assessing Future Hydrological Impacts of Climate Change on High-Mountain Central Asia: Insights from a Stochastic Soil Moisture Water Balance Model"

_EGUsphere, 2023_

## Author Comment (AC1)

**Answers to Reviewer 01 Comments**

Note: Reviewers comments in **bold**, answers in *italics* type.

**This article describes the implementation of an analytic steady-state water balance on 221 sub-catchments in central Asia. It is very well put together, reads easily from start to finish, has extensive useful figures showing the results, a careful splitting of the maths into "just the important results" in the main text and all the "tedious derivation" in the Appendix, and some thoughtful caveats in the Discussion.**

*The authors want to thank the reviewer for these kind words. We are also grateful for the thorough comments below which we individually address.*

**It is nit-picking to point out on line 169 that you only need dx(t)/dt=0 in Eq(2), as a value of w0=0 would indicate an impermeable land surface where P=Q, or a volume with no storage such that the evaporative process has no opportunity to occur.**

*This is, of course, correct and we propose changing the inline equation correspondingly to dx(t)/dt=0 in the revision.*

**Figure 8 does show very low values of w0, and linear fits crossing the x-axis, but there is no indication in the text that w0 ever reduced to zero (or negative); Zr as a component of w0 was positively correlated to temperature which always increased in the future periods used.**

*The D66 Zr root zone depth is very low in the high mountain parts of the study region, i.e., the Pamir plateau and the highest elevation parts in the Kyrgyz Naryn catchment. We propose to add this information to the main text in the paragraph starting on line 419.*

*The linear fit crossing to below zero values is misleading as it is non-physical. We thus also propose to edit the Figure 8 correspondingly to not extend the best fit linear model to the Zr<0 region.*

**Given that the solution to the water balance is a single line equation that is solved explicitly, why do the authors go down the path of using differentials to estimate changes in Q? Such an approach is usually taken when the computation is much more complex or requires some iterative solution, so for small enough perturbations the differential is a good estimate of change in output. You already need to calculate new values of E and P and all their components for the three future periods, so the new Q value could be calculated directly and compared to the baseline period. The only indication of the values (or magnitude) of the sensitivity coefficients is Figure 11 which**

**visually indicates that, in this environment, change in Q is a simple tug-of-war between increasing precipitation event depth (alpha) and increasing active soil depth (Zr), with both Ep and Lambda of little importance.**

*This is a valid point. There are different approaches to compute sensitivity of discharge to climate and different routes can be taken. This includes a) the direct computation (alluded to by the reviewer), b) the derivation of analytical climate sensitivities (approach followed in the paper), c) the intercomparison between calibrated models of many basins across a consistent geographic area, d) the empirical estimates of impacts from historic changes in a basin, and e) the establishment of empirical relationships between runoff, temperature, and precipitation for basins in comparable hydroclimatic zones[1]. Our approach was inspired by the paper by Roderick and Farquhar which analytically defined the climate sensitivity coefficients for a steady-state Budyko model[2]. We believe that this approach is best suited to identify and attribute expected changes in water availability to key hydrological model factors determining it, given the limited data availability in the region under consideration.*

*The reviewer describes the relevance of the individual sensitivity coefficients as tug-of-water between increasing alpha on the one hand and w0 (Zr) on the other, with the other sensitivity coefficients of little importance. This is a very nice way to put things and we believe that the discussion in our manuscript should be extended along these lines in discussion in Chapter 3.2 Quantification of Climate Impacts. With these results at hand, we should also more clearly point out that a 30-years climate time scale might be short when viewed from the perspective of typical time scales over which rooting depths and soils change over a larger domain. In other words, the relative importance of changes in w0 might be much more damped as, increasing the relative importance of the other sensitivity indices accordingly. We suggest adding this point also to our discussion.*

**The caption for Figure 15 is wrong. According to the map key, these are "relative changes in" precipitation frequency and depth.**

*This is correct and we suggest editing the caption of the Figure to reflect the fact that these are relative changes.*

**Figures B2-4 show the same parameter, Zr, but with three different ranges on the same continuous rainbow colour scale. They cannot be easily visually compared and require a non-linear bin-style scale that is the same for all three. Continuous colour ramps are most useful in remote sensing applications particularly, where the individual items being coloured are relatively small, the same shape and size, and there are very many of them (the false shaded DEM under the figures is a perfect example). For only 221 irregularly shaped, sized**
* * *
[1] Sankarasubramanian, Vogel, and Limbrunner, "Climate Elasticity of Streamflow in the United States."
[2] Roderick and Farquhar, "A Simple Framework for Relating Variations in Runoff to Variations in Climatic Conditions and Catchment Properties."

**and arranged polygons this is wasted. I would also suggest moving Figure B5 to follow B1 as they are the saturation and wilting points of the soil, or potentially just show a single map with their difference which is the input (s1-sw) to w0.**

*These are two very good suggestions. We thus suggest recoloring the current Figures B2-B4 accordingly so that they become comparable and to move B5 to become the new B2.*

**Perhaps the most important follow-on, and unfortunately it may no longer be a simple analytic solution, is to consider inter-annual variation (see eg. Zhang et al, 2008 using a Budyko framework). It may be possible to remain analytic on an annual basis with storage change zero over a year, but storage carryover between quarterly or monthly sub-annual intervals in a matrix equation. From the human perspective, it is not only important to know how much more flow we can expect but also when. If the water is used for irrigation of crops but will arrive in a different month in the future, then which crops are most appropriate is an issue. If most of the future flow is concentrated over a short period, then this could lead to flooding or loss of opportunity for use in irrigation or domestic water consumption. This also has implications for management with regard to storage in-stream via dams, off-stream with other engineered structures, or some form of managed aquifer recharge to mitigate flow seasonality.**

**Zhang L, Potter N, Hickel K, Zhang Y and Shao Q (2008) Water balance modeling over variable time scales based on the Budyko framework – Model development and testing. Journal of Hydrology, 360, 117-131, doi: 10.1016/j.jhydrol.2008.07.021**

*Yes, the reviewer brings up a very important point, one that we also tried to allude to in the current text in the paragraph starting on line 585. We propose to extend the discussion in the main text of the article along the lines suggested here and to also cite the suggested reference to the Zhang et al. 2008 paper. The mismatch in timing of water availability versus supplies in selected regions was already brought up by Siegfried et al. (2012) and remains an important future issue in the region and elsewhere. This is even more true under the currently observed deglaciation and the associated loss of water storage in the form of ice in high mountain Central Asia (see lines 585 and following).*

*The reason that we had decided to put our focus on changes in the mean annual water balance in the first place was the following. Out of the total of 299 station throughout the region that we could localize, time series discharge data from only 135 gauging stations was available, i.e., from less than half of the entire set of stations. Hence, focusing only on these, we would have had to abandon the truly regional approach pursued here and work with much less data to understand climate impacts on the seasonality of rivers. We believe however that our study can serve as a first diagnostic for providing estimates of changes in the mean water balance and then guide more detailed studies on a per subcatchment basis.*

We hope that the replies to the reviewer's comments/suggestions are satisfactory to her/him.

Kind regards, Tobias Siegfried (on behalf of all the co-authors).

---

## Author Comment (AC2)

**Answers to Reviewer 02 Comments**

Note: Reviewers comments in **bold**, answers in *italics* type.

**The manuscript is very well written and easy to follow.**

*The authors thank the reviewer for these encouraging words. The authors acknowledge the fact that this regional study relies on a simplified understanding of the complex hydrological processes in the topographically highly diverse region. We believe that the approach followed here is warranted given the regional scope of this impact study which relies on data that we had put together of 221 catchments throughout the region covering an area of more than 400'000 km². Such a unified view of this highly fragile yet fascinating world region has never been carried out before and we believe that our thorough study contributes in an essential manner to better prepare local stakeholders and populations for future changes.*

*We are sure that the detailed answers below to observations and questions from the reviewer help to address his/her skepticism towards the approach.*

**A very simplified water balance model is applied with numerous assumptions that require more discussions and comparisons with data sets like:**

- **Evaporation (including transpiration) depends only on potential evaporation, what about water availability?**

- **The partitioning of runoff and evaporation is it dependent on the topography?**

- **Parameters k1 and k2 (equation (17)) are set to 100 and 5. In Oudin et al. (2005) quoted by the authors, these parameters have to be calibrated with values depending on the hydrological model... Values of 100 and 5 are based on the water catchments they studied, which are different from the catchments addressed here.**

- **Concerning the rooting depth, it is assumed that data from Fan et al. (2017) are a good compromise, but it is not explained why ....**

- **Equation (18) is an oversimplified model that needs to be checked versus data.**

**The authors mentioned most of these simplifications.**

*The model used here is a well-established stochastic soil moisture dynamics model. As in the main text, we refer to it as PSM model. Good, comprehensive overviews of the PSM*

*model and its applications globally are available[1]. The PSM model has also been applied in arid and semi-arid contexts. We would like to clarify the raised issues above one by one.*

- *Lines 161 and following explain the dependence of actual evaporation on potential evaporation. We clearly state that actual evaporation depends on effective relative soil moisture x(t) and thus on water availability in each of the studied subcatchments in the PSM model.*
- *The partitioning of available water into runoff and evaporation is shown in Equations (4) – (6) and the difference between the PSM model and the Budyko model is highlighted in Equations (7) and (8). The PSM model main variables are the basin aridity index ϕ and the basin storage index γ. As for the aridity index ϕ, it depends on the basin potential evaporation which itself depends on temperature. The dependence of ϕ on topography thus enters via the temperature lapse rate. As for γ, the dependence on topography comes from reduced rooting depth at high altitudes.*
- *We totally agree with the reviewer that these parameters are generic and not necessarily suitable for the catchments under consideration. In fact, we show the bias of potential evaporation computed using k1 = 100 and k2 = =5 in the left plate in Figure 5. We therefore use a bias correction method to harmonize these Ep values with the cones computed via Penman in the Chelsa V21 dataset. The reviewer is kindly requested to check the discussion starting on lines 338 – 353.*
- *In relation to soil data, we spent a considerable amount of time to access and compare different global datasets. This is evidenced in Section 2.5 in the article. For rooting depth, we compared three global datasets (see lines 356 – 357). We believe that the data from Fan et al., 2017 provides a good compromise product when compared with the other products as it is neither biased to the low nor to the high ends in the spectrum of available data (lines 365 – 368 and Figures B2 – B4).*
- *We agree that the model presented in Equation (18) is a simple linear model relating rooting depth to temperature (see also comment above). The data for fitting the linear model are shown in the left plate of Figure 8 that shows the clear dependence of the rooting depth on mean temperatures. These data are used for fitting the linear model. We hope that the discussion in the main text is sufficient in motivating our choice of the model.*

**Because these simplifications are very strong and sometimes not consistent with our knowledge, they must be checked by comparisons with existing data (at least, at the scale of one water catchment). Without that checking, there is no possibility to verify the plausibility of the results and the paper appears like a modelling exercise not suitable for HESS.**

*We thank the reviewer for sharing these concerns. We would like to point out that we spent a lot of time collecting a consistent dataset for the entire region. The resulting manuscript*
* * *
[1] Porporato and Yin, *Ecohydrology: Dynamics of Life and Water in the Critical Zone*; Rodríguez-Iturbe and Porporato, *Ecohydrology of Water-Controlled Ecosystems: Soil Moisture and Plant Dynamics*.

*draft by Marti, B. S., Yakovlev, A., Karger, D. N., Ragettli, S., Zhumabaev, A., Wakil, and Siegfried, T.: Geo-Located Discharge Time Series for Mountainous Rivers in Central Asia: A Novel Dataset for Water Balance Modelling and Runoff Forecasting, Rev., 2023 has just been accepted for publication in the journal Nature Scientific Data. This is the first time such dataset has been put together and used in a comprehensive way for a hydro-climatological impact assessment of global change. HESS is well known to public important contributions in this field, and we believe that our contribution adds in a targeted manner to ongoing discussions in this field.*

We hope that the replies to the reviewer's comments/suggestions are satisfactory to her/him.

Kind regards, Tobias Siegfried (on behalf of all the co-authors).

---

## Author Comment (AC3)

**Answers to Reviewer 03 Comments**

Note: Reviewers comments in **bold**, answers in *italics* type.

**This study evaluate the hydrological impact on the central Aisa. The results are interesting. However, this study make many assumption to simplify the water balance model, which need extra explanation and work to make sure the results are accurate.**

*We thank the reviewer for his kind words in relation to the results of this study. It goes without saying that we totally agree with the reviewer that the simplified stochastic soil moisture dynamics model (PSM model) utilized in our studied makes many assumptions. The model is well-established in literature[1]. We do not think that there is value added in repeating the underlying assumptions of the PSM model in detail in our manuscript as it is an application paper of that model.*

**Line 140, water balance equation. this equation is only for closed system. what happens between the water interchaning between subcatchements and glacier melt? Please use the term evapotranspiration to represent both transpiration and evaporation.**

*The PSM model specifies the partitioning of available water into runoff and evaporation at the level of the subcatchments. The water balance of the 221 subcatchments is modelled independently from each other, i.e., other is no routing between catchments as we study the effects at the level of the individual subcatchments. This allows us to then look at the statistics over the larger basins such as Issy Kul, Chu River, Talas River, Syr Darya, and Amu Darya. As for glacier melt, we take this into account in a separate manner as described in detail in Chapter 2.6.*

*In relation to the debate about evaporation and evapotranspiration, we adhere to the definition given by Miralles et al., 2020[2]. Therefore, as explained on line 142 and following, our definition of evaporation encompasses evaporation from inside leaves (transpiration), evaporation from bare soils, evaporation from intercepted precipitation, evaporation from open water surfaces, and finally, evaporation over ice- and snow-covered surfaces. We believe that the existing explanation in the text is sufficient and are sure that the reviewer agrees with us on this.*

**Line 165, evapotranspiration takes up a large poration of the water balance. This study made assumption that Em is not dependent on time. However, the E could vary a lot seasonally, e.g. serveral times higher in summer than winter.**
* * *
[1] Porporato and Yin, *Ecohydrology: Dynamics of Life and Water in the Critical Zone*; Rodríguez-Iturbe and Porporato, *Ecohydrology of Water-Controlled Ecosystems: Soil Moisture and Plant Dynamics*.
[2] Miralles et al., "On the Use of the Term 'Evapotranspiration.'"

**Besides, the Em is also dependent on the vegetation type, soil moisture (another important factor in this study) and temperature. This study project 3 degree climate change which could significantly impact the Em term. How do you make sure this assumption will not affect the final result? I recommend to re-model the evapotranspiration term. Please refer study: Zhou, Z., & Guo, Q. (2022). Drainage alternatives for rain gardens on subsoil of low permeability: Balance among ponding time, soil moisture, and runoff reduction.** *Journal of Sustainable Water in the Built Environment*, *8*(3), 05022002.

*We thank the reviewer for this comment that consists of several elements. First, let us talk about the comment about the seasonal variability of the fluxes under consideration. We completely agree that the model presented here cannot resolve subannual variability as it computes a mean annual flow partitioning at the subcatchments scale. This applies to Em (as mentioned by the reviewer) but also to all other fluxes. Our focus on mean annual fluxes is motivated by the fact that we wanted to keep as much as possible a holistic approach of the high-mountain Central Asia region and not only limit ourselves to the 135 gauging stations (less than half of the total set of 299 stations) for which we could obtain time series data and thus study seasonal discharge in detail.*

*We are confident, however, that our study helps to identify priority basins and subcatchments for which more detailed climate impact studies can be carried out. These detail studies could then also use a different modeling approach (see also Reviewer 1 comments and our reply) where impacts on seasonality changes can be modelled. These can include increasing cold season discharge, shift of discharge peak towards spring, and reduction of summer discharge peak, etc.*

*Second, we are not sure that we understand the reviewer correctly when he/she mentions that it is not clear how a 3-degree climate change impacts the Em term. We have described how we compute the impact of climate change on the potential evaporation Ep in Equation 17 (see lines 338 – 353). While hoping that we have not misunderstood the reviewer, we believe that the description in the manuscript is sufficient.*

*We are very grateful for the reviewer's suggestion to cite the interesting study by Zhou et al. (2022). We think, however, that the reference is not exactly relevant for our paper as the Zhou et al. (2022) study neither uses a PSM- or Budyko-type model and does not focus on the Central Asia regions. Furthermore, our study has nothing to do with rain gardens and drainage systems in the United States geographically speaking. We hope that the reviewer understands this.*

We hope that the replies to the reviewer's comments/suggestions are satisfactory to her/him.

Kind regards, Tobias Siegfried (on behalf of all the co-authors).